# Benchmarking Offline Reinforcement Learning on Real-Robot Hardware

**Nico Gürtler**[1], **Sebastian Blaes**[1], **Pavel Kolev**[1], **Felix Widmaier**[1], **Manuel Wüthrich**[2],
**Stefan Bauer**[3], **Bernhard Schölkopf**[1], and **Georg Martius**[1]

[1]Max Planck Institute for Intelligent Systems*
[2]Harvard University
[3]KTH Stockholm

## Abstract

Learning policies from previously recorded data is a promising direction for real-world robotics tasks, as online learning is often infeasible. Dexterous manipulation in particular remains an open problem in its general form. The combination of offline reinforcement learning with large diverse datasets, however, has the potential to lead to a breakthrough in this challenging domain analogously to the rapid progress made in supervised learning in recent years. To coordinate the efforts of the research community toward tackling this problem, we propose a benchmark including: i) a large collection of data for offline learning from a dexterous manipulation platform on two tasks, obtained with capable RL agents trained in simulation; ii) the option to execute learned policies on a real-world robotic system and a simulation for efficient debugging. We evaluate prominent open-sourced offline reinforcement learning algorithms on the datasets and provide a reproducible experimental setup for offline reinforcement learning on real systems. Visit `https://sites.google.com/view/benchmarking-offline-rl-real` for more details.

## 1 Introduction

Reinforcement learning (RL) (Sutton et al., 1998) holds great potential for robotic manipulation and other real-world decision-making problems as it can solve tasks autonomously by learning from interactions with the environment. When data can be collected during learning, RL in combination with high-capacity function approximators can solve challenging high-dimensional problems (Mnih et al., 2015; Lillicrap et al., 2016; Silver et al., 2017; Berner et al., 2019). However, in many cases online learning is not feasible because collecting a large amount of experience with a partially trained policy is either prohibitively expensive or unsafe (Dulac-Arnold et al., 2020). Examples include autonomous driving, where suboptimal policies can lead to accidents, robotic applications where the hardware is likely to get damaged without additional safety mechanisms, and collaborative robotic scenarios where humans are at risk of being harmed.

Offline reinforcement learning (offline RL or batch RL) (Lange et al., 2012) tackles this problem by learning a policy from prerecorded data generated by experts or handcrafted controllers respecting the system's constraints. Independently of how the data is collected, it is essential to make the best possible use of it and to design algorithms that improve performance with the increase of available data. This property has led to unexpected generalization in computer vision (Krizhevsky et al., 2012; He et al., 2016; Redmon et al., 2016) and natural language tasks (Floridi & Chiriatti, 2020; Devlin et al., 2018) when massive datasets are employed. With the motivation to learn similarly capable decision-making systems from data, the field of offline RL has gained considerable attention. Progress is currently measured by benchmarking algorithms on simulated domains, both in terms of data collection and evaluation.

---

*Correspondence to `nico.guertler@tuebingen.mpg.de`

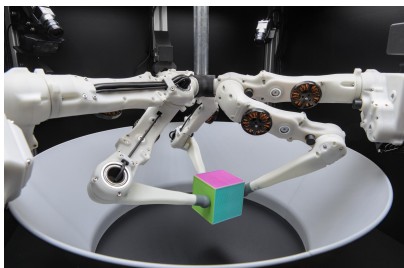 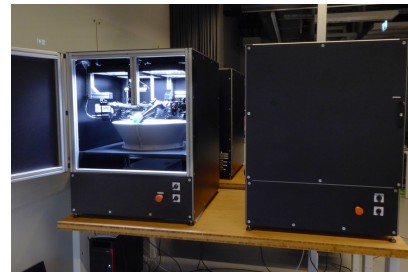

Figure 1: **The TriFinger manipulation platform (Wüthrich et al., 2021; Bauer et al., 2022).** Left: The robot has 3 arms with 3 DoF each. The cube is constrained by a bowl-shaped arena, allowing for unattended data collection. Right: A cluster of these robots for parallel data collection and evaluation.

Yet, real-world data differs from simulated data qualitatively and quantitatively in several aspects (Dulac-Arnold et al., 2020). First, observations are noisy and sometimes faulty. Second, real-world systems introduce delays in the sensor readings and often have different sampling rates for different modalities. Third, the action execution can also be delayed and can get quantized by low-level hardware constraints. Fourth, real-world environments are rarely stationary. For instance, in autonomous robotics, battery voltages might drop, leading to reduced motor torques for the same control command. Similarly, thermal effects change sensor readings and motor responses. Abrasion changes friction behavior and dust particles can change object appearances and sensing in general. Fifth, contacts are crucial for robotic manipulation but are only insufficiently modeled in current physics simulations, in particular for soft materials. Lastly, physical robots have individual variations.

Since real-world data is different from simulated data, it is important to put offline RL algorithms to the test on real systems. We propose challenging robotic manipulation datasets recorded on real robots for two tasks: object pushing and object lifting with reorientation on the TriFinger platform (Wüthrich et al., 2021). To study the differences between real and simulated environments, we also provide datasets collected in simulation. Our benchmark of state-of-the-art offline RL algorithms on these datasets reveals that they are able to solve the moderately difficult pushing task while their performance on the more challenging lifting task leaves room for improvement. In particular, there is a much larger gap between the performance of the expert policy and offline-learned policies on the real system compared to the simulated system. This underpins the importance of real-world benchmarks for offline RL. We furthermore study the impact of adding suboptimal trajectories to expert data and find that all algorithms are 'distracted' by them, i.e., their success rate drops significantly. This identifies an important open challenge for the offline RL community: robustness to suboptimal trajectories.

Importantly, a cluster of TriFinger robots is set up for evaluation of offline-learned policies for which remote access can be requested for research purposes. With our dataset and evaluation platform, we therefore aim to provide a breeding ground for future offline RL algorithms.

## 2 THE TRIFINGER PLATFORM

We use a robot cluster that was initially developed and build for the *Real Robot Challenge* in 2020 and 2021 (Bauer et al., 2022). The robots that constitute the cluster are an industrial-grade adaptation of a robotic platform called TriFinger, an open-source hardware and software design introduced in Wüthrich et al. (2021), see Fig. 1.The robots have three arms mounted at a 120 degrees radially symmetric arrangement with 3 degrees of freedom (DoF) each. The arms are actuated by outrunner brushless motors with a 1:9 belt-drive, yielding high agility, low friction, and good force feedback (details on the actuator modules can be found in Grimminger et al. (2020)). Pressure sensors inside the elastic fingertips provide basic tactile feedback. The working area, where objects can be manipulated, is encapsulated by a high barrier to ensure that the object stays inside the arena even during aggressive motions. This is essential for operation without human supervision. The robot is inside a closed housing that is well lit by top-mounted LED panels making the images taken by three high-speed global shutter cameras consistent. The cameras are distributed between the arms to ensure objects are always seen by any camera. We study dexterous manipulation of a cube whose pose is estimated by a visual tracking system with 10 Hz.

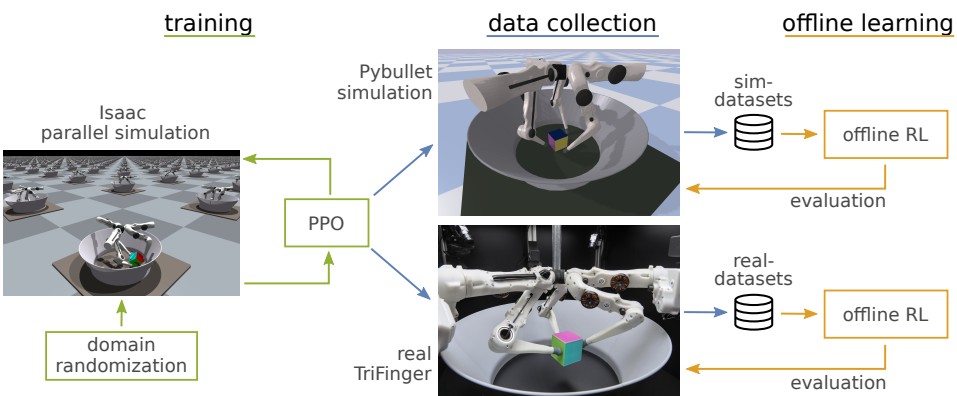

Figure 2: **Overview of our approach.** Policies are trained with domain randomization in a parallel simulation using Isaac Gym (Makoviychuk et al., 2021) and then deployed in the PyBullet (Coumans & Bai, 2016) simulation and on the real system without fine-tuning to collect the datasets. We train state-of-the-art offline RL algorithms on these datasets and evaluate them on the respective system, i.e, the simulator or the real-robot cluster.

To abstract away the low-level details, we developed a software with a simple Gym (Brockman et al., 2016) interface in Python that can interact with the robots at a maximal rate of 1 kHz in position control or torque control mode (see (Wüthrich et al., 2021) for details). We use a control frequency of 50 Hz and torque control for this work. We have a custom object tracking tool to provide position and orientation of a single colored cube in the environment, allowing algorithms to work without visual input.

On top of this interface, we have developed a submission system that allows users to submit *jobs* to a cluster of these robots for unattended remote execution. This setup was specifically adapted for ease of use in the offline RL setting and was extensively tested. We will provide researchers with access to the robot cluster, which will allow them to evaluate the policies they trained on the dataset proposed herein. To ease development and study the fundamental differences between simulated and real world data, we provide a corresponding simulated environment using PyBullet (Coumans & Bai, 2016).

To summarize, the hardware and software have the following three key properties: i) physically capable of dexterous manipulation; ii) robust enough for running and evaluating learning methods, and iii) easy to use (robot hardware and simulator) and integrated in existing code frameworks.

## 3 THE TRIFINGER DATASETS

We consider two tasks that involve a colored cube: pushing the cube to a target location and lifting the cube to a desired location and orientation. To create behavioral datasets for these tasks on the TriFinger platform, we need expert policies, which we obtain using reinforcement learning in a parallel simulation environment with domain randomization. Fig. 2 visualizes the entire procedure – from training to data collection to offline learning. In this section, we describe the tasks and the data collection while Sec. 4 is dedicated to benchmarking offline RL algorithms on the collected data.

### 3.1 DEXTEROUS MANIPULATION TASKS

We consider two tasks on the TriFinger platform that require dexterous manipulation of a cube:

**Push** The goal is to move the cube to a target location. This task does not require the agent to align the orientation of the cube; the reward is based only on the desired and the achieved position.

**Lift** The cube has to be picked up and moved to a target pose in the air which includes position and orientation. This requires flipping the cube on the ground, obtaining a stable grasp, lifting it to a target location and rotating it to match the target orientation.

Following prior work (Hwangbo et al., 2019; Allshire et al., 2022) we define the reward by applying a logistic kernel $k(x) = (b + 2) \left(\exp(a\|x\|) + b + \exp(-a\|x\|)\right)^{-1}$ to the difference between desired and achieved position (for the Push task) or the desired and achieved corner points of the cube (for

the Lift task). The parameters $a$ and $b$ control the length scale over which the reward decays and how sensitive it is for small distances $x$, respectively. This yields a smooth, dense and bounded reward, see Appendix B.3 for details. We define success in the pushing task as reaching the goal position with a tolerance of 2 cm. For the lifting task we additionally require to not deviate more than 22 degrees from the goal orientation.

We note that pushing is more challenging than it may appear due to inelastic collisions between the soft fingertips and the cube which are not modeled in rigid body physics simulators. Furthermore, the performance of policies can be quite sensitive to the value of sliding friction. Lifting is, however, even more challenging as flipping the cube based on a noisy object-pose with a low refresh rate is error-prone, learning the right sequence of behaviors requires long-term credit assignment and dropping the cube often results in loosing all progress in an episode.

## 3.2 TRAINING EXPERT POLICIES

We train expert policies with online RL in simulation which we then use for data collection on the real system (Fig. 2). We build on prior work which achieved sim-to-real transfer for a dexterous manipulation task on the TriFinger platform (Allshire et al., 2022). This approach replicates the real system in a fast, GPU-accelerated rigid body physics simulator (Makoviychuk et al., 2021) and trains with an optimized implementation (Makoviichuk & Makoviychuk, 2022) of Proximal Policy Optimization (Schulman et al., 2017) with a high number of parallel actors. We furthermore adopt their choice of a control frequency of 50 Hz and use torque control to enable direct control of the fingers. The sensor input is the proprioceptive sensor information and the object pose. In order to obtain policies that are robust enough to work on the real system, domain randomization (Tobin et al., 2017; Mandlekar et al., 2017; Peng et al., 2018) is applied: for each episode, physics parameters like friction coefficients are sampled from a distribution that is likely to contain the parameters of the real environment. Additionally, noise is added to the observations and actions to account for sensor noise and the stochasticity of the real robot. Furthermore, random forces are applied to the cube to obtain a policy that is robust against perturbations, as in (Andrychowicz et al., 2020). The object and goal poses are represented by keypoints, i.e., the Cartesian coordinates of the corners of the cube. This choice was empirically shown to accelerate training compared to separately encoding position and orientation in Cartesian coordinates and a quaternion (Allshire et al., 2022).

To improve the robustness of the trained policies across the different robot instances and against other real-world effects like tracking errors due to accumulating dust, we modified the implementation of Allshire et al. (2022) in several ways. While the original code correctly models the 10 Hz refresh rate of the object pose estimate, we found it beneficial to also simulate the delay between the time when the camera images are captured and when they are provided to the agent. This delay typically ranges between 100 ms and 200 ms. We furthermore fully randomize the initial orientation of the cube and use a hand-crafted convex decomposition of the barrier collision mesh to avoid artifacts of the automatic decomposition, to which the policy can overfit. For the pushing task we penalized rapid changes in the cube position and orientation as sliding and flipping the cube transfers less well to the real system than moving it in a slow and controlled manner. We observed that policies trained with RL in simulation tend to output oscillatory actions, which does not transfer well to the real system and causes stronger wear effects (Mysore et al., 2021). To avoid this, we penalized changing the action which led to smoother movements and better performance. For the Lift task we additionally consider a policy which was trained with an Exponential Moving Average on the actions as we observed that vibrations on the real robot can lead to slipping and dropping (see Fig. S14 (b)). These vibrations might be caused by elastic deformations of the robot hardware and the complex contact dynamics between the soft fingertips and the cube that are not modeled in simulation. As also observed in Wang et al. (2022), the performance on the real system varies significantly with training seeds. We evaluated 20 seeds on the real system and used the best one for data collection. More details on training in simulation can be found in the Appendix C.1.

## 3.3 DATA COLLECTION

We collected data for the pushing and lifting tasks both in a PyBullet (Coumans & Bai, 2016) simulation (Joshi et al., 2020) and on the real system, as shown in Fig. 2. To ensure that the data collection procedures are identical, we run the same code with the simulator backend and with the

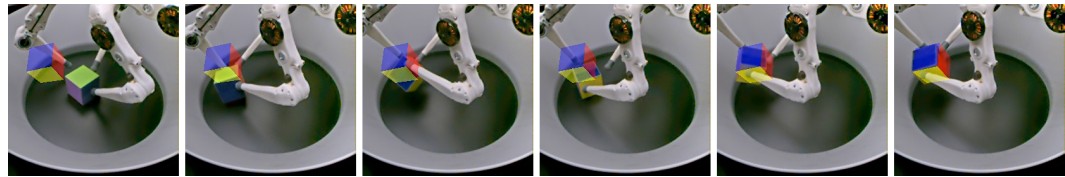

Figure 3: **An example behavior of our expert policy on the lifting task.** The robot is grasping and reorienting the object to reach the desired target location and orientation (transparent cube).

real-robot backend on six TriFinger platforms (Fig. 1). We observed that policies learned in simulation struggle with retrieving the cube from the barrier on the real robot[1]. To alleviate this problem, we employ a predefined procedure to push the cube away from the barrier between episodes. The resulting starting positions together with the target positions are visualized in Fig. S12. During data collection, self-tests are performed at regular intervals to ensure that all robots are fully functional.

For the dataset to be useful to the community, it should be possible to evaluate offline-learned policies on the real robots. As machine learning models can be computationally demanding, we wait for a fixed time interval between receiving a new observation and starting to apply the action based on this observation. This time budget is allocated for running the policy without deviating from the control frequency used for data collection. We choose 10 ms for the Push task and 2 ms for the Lift task as we found training with bigger delays difficult. Note that our expert policies run in less than 1 ms.

We provide as much information about the system as possible in the observations. The robot state is captured by joint angles, angular velocities, recorded torques, fingertip forces, fingertip positions, fingertip velocities (both obtained via forward kinematics), and the ID of the robot. The object pose is represented by a position, a quaternion encoding orientation, the keypoints, the delay of the camera images for tracking and the tracking confidence. The observation additionally contains the last action, which is applied during the fixed delay between observation and action. The desired and achieved goal are also included in the observation and contain either a position for the Push task or keypoints for the Lift task. Some observations provide redundant information; however, we believe this simplifies working with the datasets, as we provide user-friendly Python code that implements filtering as well as automatically converting observations to a flat array. Moreover, we additionally publish a version of each dataset with camera images from three viewpoints. We believe that directly learning from these image datasets is an exciting challenge for the community[2] and that they are moreover valuable in their own right as a large collection of robotic manipulation video footage.

For each task we consider pure expert data (Expert), mixed data recorded with a range of training checkpoints (Mixed), a combination of 50% expert trajectories and 50% trajectories recorded with a weaker policy with additive Gaussian noise on the actions (Weak&Expert) and the 50% expert data in Weak&Expert (Half-Expert). We run the same policies in simulation and on the real system. An exemplary expert behavior for the Lift task is shown in Fig. 3. Episodes last for 15 s for the Push task and 30 s for the Lift task as reorienting, grasping and lifting the cube requires more time. For the Push task, we collect 16 h of interaction for each dataset, corresponding to 3840 episodes and 2.8 million transitions. For the more demanding Lift task, we collect 20 h of robot interactions corresponding to 2400 episodes and 3.6 million transitions. The Half-Expert datasets contain the expert data that is included in the corresponding Weak&Expert datasets to isolate the effects of reducing the amount of expert data and adding suboptimal trajectories. The average success rates are provided next to the offline learning results in Table 1 and 2 (see *data* column). In Appendix B, we give a detailed analysis along with a specification (Table S4) and statistics (Table S5) of the offline RL datasets.

## 4    BENCHMARKING OFFLINE REINFORCEMENT LEARNING ALGORITHMS

We benchmark offline RL algorithms on pairs of simulated and real datasets and study the impact of data quality on their performance. We limit our evaluation to the best algorithms provided in the

---

[1]Possibly because the object tracking performance is slightly worse at the barrier or the dynamics of the outstretched fingers aligns less well between simulation and reality.

[2]At the time of writing we cannot benchmark on these datasets since the robot cluster does not provide GPU-access at the moment. This will likely change in the near future.

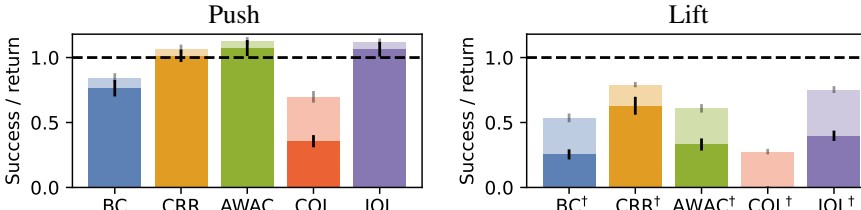

Figure 4: **Average normalized success rates and returns (light colors) on real robots**. Each quantity is normalized by the dataset mean and averaged over all Push or Lift datasets.

Table 1: **Pushing: Success rate on the TriFinger-Push Datasets.** 'data' denotes the mean over the dataset. Average and standard deviation over five training seeds. A star $*$ indicates significance w.r.t. all other methods using Welch's $t$-test with $p < 0.05$.

| Push-Datasets | data | BC | CRR | AWAC | CQL | IQL |
|---|---|---|---|---|---|---|
| Sim-Expert | 0.95 | $0.83 \pm 0.02$ | $\mathbf{0.94 \pm 0.04}$ | $0.92 \pm 0.03$ | $0.03 \pm 0.01$ | $0.88 \pm 0.04$ |
| Sim-Half-Expert | 0.95 | $0.71 \pm 0.05$ | $\mathbf{0.79 \pm 0.05}$ | $\mathbf{0.79 \pm 0.02}$ | $0.05 \pm 0.02$ | $0.70 \pm 0.06$ |
| Sim-Weak&Expert | 0.53 | $0.53 \pm 0.09$ | $\mathbf{0.88 \pm 0.03}$ | $0.83 \pm 0.05$ | $0.17 \pm 0.03$ | $0.66 \pm 0.14$ |
| Sim-Mixed | 0.76 | $0.53 \pm 0.04$ | $0.09 \pm 0.10$ | $\mathbf{0.84 \pm 0.06}^*$ | $0.02 \pm 0.01$ | $0.69 \pm 0.07$ |
| Real-Expert | 0.92 | $0.74 \pm 0.05$ | $\mathbf{0.87 \pm 0.07}$ | $0.80 \pm 0.03$ | $0.54 \pm 0.13$ | $0.75 \pm 0.08$ |
| Real-Half-Expert | 0.92 | $0.66 \pm 0.08$ | $\mathbf{0.78 \pm 0.04}$ | $0.76 \pm 0.10$ | $0.48 \pm 0.08$ | $0.70 \pm 0.08$ |
| Real-Weak&Expert | 0.51 | $0.48 \pm 0.10$ | $\mathbf{0.84 \pm 0.06}^*$ | $0.69 \pm 0.06$ | $0.14 \pm 0.04$ | $0.68 \pm 0.05$ |
| Real-Mixed | 0.49 | $0.29 \pm 0.06$ | $0.30 \pm 0.06$ | $0.61 \pm 0.09$ | $0.02 \pm 0.02$ | $\mathbf{0.66 \pm 0.08}$ |

open-source library d3rlpy (Seno & Imai, 2021) to keep the required robot time for the evaluation of the trained policies manageable. Namely, we benchmark the following algorithms: BC (Bain & Sammut, 1995; Pomerleau, 1991; Ross et al., 2011; Torabi et al., 2018), CRR (Wang et al., 2020), AWAC (Nair et al., 2020), CQL (Kumar et al., 2020), and IQL (Kostrikov et al., 2021b). We report results for two sets of hyperparameters: The default values (except for CQL for which we performed a grid search on Push-Sim-Expert as the default parameters did not learn) and the results of a grid search on Lift-Sim-Weak&Expert (marked by $^\dagger$). Details about the hyperparamters and their optimization are in Appendix C.3. We think that the performance at the default hyperparameters is highly relevant for offline RL on real data as optimizing the hyperparameters without a simulator is often infeasible.

## 4.1 RESULTS

We train with five different seeds for each algorithm and evaluate with a fixed set of randomly sampled goals. Details about the policy evaluation on the simulated and real set-up can be found in Appendix D. We report success rates in the main text and returns in Appendix A.

The benchmarking results for the Push task are summarized in Table 1. Most of the offline RL algorithms perform well but the performance on the real data is generally worse. An exception to this pattern is CQL which performs poorly on the simulated Push task and gains some performance on the real data, perhaps due to the broader data distribution of the stochastic real-world environment. As expected BC performs well on the expert datasets but cannot exceed the dataset success rate on the Weak&Expert data, unlike CRR, AWAC, and IQL. On the expert data, CRR and AWAC match the performance of the expert policy in simulation but fall slightly behind on the real data. Interestingly, the performance of CRR is not negatively impacted by weak trajectories (e.g. 84% on Real-Weak&Expert compared to 78% on Real-Half-Expert, where the latter only contains the expert data portion). We provide insights into how the behavior policy compares to an offline-trained policy for the Push task in Fig. 6 (b) and (c).

The more challenging Lift task separates the algorithms more clearly as summarized in Table 2. CQL does not reach a non-zero success rate at all, despite our best efforts to optimize the hyperparameters (see Appendix C.3). This is in line with the results reported in Mandlekar et al. (2021) where CQL failed to learn on the datasets corresponding to more complex manipulation tasks. Kumar et al. (2021) furthermore discusses the sensitivity of CQL to the choice of hyperparameters, especially on robotic data. While the best algorithms come close to matching the performance of the expert on Lift-Sim-Expert, they fall short of reaching the dataset success rate on the real-robot data (Lift-Real-Expert).

Table 2: **Lifting: Success rate on the TriFinger-Lift Datasets.** 'data' denotes the mean over the dataset. Average and standard deviation over five training seeds. Experiments with hyperparameters optimized on Sim-Weak&Expert are marked with a $^\dagger$. Stars $^*$ and $^{**}$ indicate significance w.r.t. second best Welch's $t$-test with $p < 0.05$ and $p < 0.01$, respectively.

| Lift-Datasets | data | BC | CRR | AWAC | CQL | IQL |
|---|---|---|---|---|---|---|
| Sim-Expert | 0.87 | $0.64 \pm 0.00$ | $\mathbf{0.80 \pm 0.03}$ | $0.75 \pm 0.04$ | $0.00 \pm 0.00$ | $0.47 \pm 0.06$ |
| Sim-Half-Expert | 0.88 | $0.64 \pm 0.02$ | $\mathbf{0.78 \pm 0.02}^*$ | $0.69 \pm 0.05$ | $0.00 \pm 0.00$ | $0.04 \pm 0.01$ |
| Sim-Weak&Expert | 0.5 | $0.16 \pm 0.04$ | $0.43 \pm 0.35$ | $\mathbf{0.55 \pm 0.09}$ | $0.00 \pm 0.00$ | $0.24 \pm 0.05$ |
| Sim-Mixed | 0.68 | $0.01 \pm 0.01$ | $0.19 \pm 0.06$ | $\mathbf{0.41 \pm 0.09}^{**}$ | $0.00 \pm 0.00$ | $0.00 \pm 0.00$ |
| Sim-Expert$^\dagger$ | 0.87 | $0.80 \pm 0.04$ | $\mathbf{0.84 \pm 0.01}$ | $\mathbf{0.84 \pm 0.01}$ | $0.01 \pm 0.01$ | $0.80 \pm 0.03$ |
| Sim-Half-Expert$^\dagger$ | 0.88 | $0.67 \pm 0.09$ | $0.75 \pm 0.08$ | $\mathbf{0.82 \pm 0.02}$ | $0.00 \pm 0.01$ | $0.75 \pm 0.05$ |
| Sim-Weak&Expert$^\dagger$ | 0.5 | $0.47 \pm 0.03$ | $\mathbf{0.69 \pm 0.04}$ | $0.54 \pm 0.03$ | $0.00 \pm 0.00$ | $0.64 \pm 0.05$ |
| Sim-Mixed$^\dagger$ | 0.68 | $0.01 \pm 0.00$ | $0.20 \pm 0.07$ | $\mathbf{0.32 \pm 0.04}^*$ | $0.00 \pm 0.00$ | $0.01 \pm 0.01$ |
| Real-Smooth-Expert | 0.64 | $0.26 \pm 0.09$ | $\mathbf{0.52 \pm 0.12}^*$ | $0.34 \pm 0.04$ | $0.00 \pm 0.00$ | $0.24 \pm 0.03$ |
| Real-Expert | 0.66 | $0.27 \pm 0.09$ | $\mathbf{0.57 \pm 0.07}^{**}$ | $0.24 \pm 0.04$ | $0.00 \pm 0.00$ | $0.29 \pm 0.09$ |
| Real-Half-Expert | 0.68 | $0.15 \pm 0.01$ | $\mathbf{0.38 \pm 0.08}^{**}$ | $0.12 \pm 0.08$ | $0.00 \pm 0.00$ | $0.12 \pm 0.06$ |
| Real-Weak&Expert | 0.40 | $0.02 \pm 0.04$ | $\mathbf{0.17 \pm 0.09}$ | $0.03 \pm 0.06$ | $0.00 \pm 0.00$ | $0.11 \pm 0.13$ |
| Real-Mixed | 0.42 | $0.00 \pm 0.01$ | $\mathbf{0.18 \pm 0.07}^{**}$ | $0.02 \pm 0.01$ | $0.00 \pm 0.00$ | $0.03 \pm 0.02$ |
| Real-Expert$^\dagger$ | 0.66 | $0.28 \pm 0.04$ | $\mathbf{0.54 \pm 0.09}$ | $0.31 \pm 0.04$ | $0.00 \pm 0.00$ | $0.48 \pm 0.07$ |
| Real-Half-Expert$^\dagger$ | 0.68 | $0.25 \pm 0.07$ | $\mathbf{0.37 \pm 0.09}$ | $0.36 \pm 0.06$ | $0.01 \pm 0.01$ | $0.30 \pm 0.06$ |
| Real-Weak&Expert$^\dagger$ | 0.40 | $0.09 \pm 0.04$ | $\mathbf{0.29 \pm 0.07}^*$ | $0.12 \pm 0.06$ | $0.00 \pm 0.00$ | $0.15 \pm 0.03$ |
| Real-Mixed$^\dagger$ | 0.42 | $0.00 \pm 0.00$ | $\mathbf{0.18 \pm 0.04}^{***}$ | $0.01 \pm 0.01$ | $0.00 \pm 0.00$ | $0.02 \pm 0.01$ |

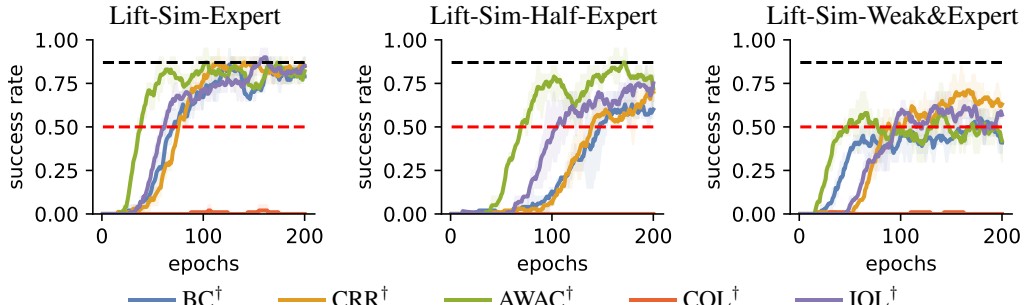

Figure 5: **Success rates during offline RL training on the simulated Lift task.** The black dashed line indicates the performance of the expert dataset, and the red dashed line depicts average success rate of the Weak&Expert dataset. Shaded areas indicate the interval between the 0.25 and 0.75 quantiles. Learning curves for all experiments are shown in Appendix A.6.

On the Real-Smooth-Expert dataset the success rates are, on average, slightly lower, probably due to the non-Markovian behavior policy. The returns are generally higher, however, likely due to the smoothed expert reaching higher returns (see Table S2).

On the Lift-Weak&Expert datasets all algorithms perform significantly worse than on the expert data. This effect is most pronounced on the real robot data and calls the ability of the algorithms to make good use of the 50% expert trajectories into question. To verify that the performance drop is not due to only half of the expert data being available, we also train solely on the expert trajectories contained in the Weak&Expert dataset. We refer to this dataset as Half-Expert. We find that the resulting policy performs significantly better, ruling out that the performance drop on the Lift-Weak&Expert dataset is exclusively caused by a lack of data. Fig. 5 shows that this is true for the simulated Lift datasets as well. We conclude that the benchmarked offline RL algorithms still exhibit BC-like behavior, i.e., they are distracted by suboptimal data. Their poor performance on the Mixed datasets further underpins their reliance on imitation of a consistent expert as all algorithms struggle to learn from the data collected by a range of training checkpoints (see tables 1 and 2 and learning curves in section A.6).

The Lift datasets, in particular those recorded on the real robots, generally contain more useful sub-trajectories than apparent from the success rate, which is defined as achieving the goal pose *at the end* of an episode. For example, during 87% of all episodes in the Lift-Real-Expert dataset the goal pose is achieved *at some point* but the best algorithm CRR achieves a success rate of only 57%. This suggests that these datasets have an untapped potential for trajectory stitching. As the observations

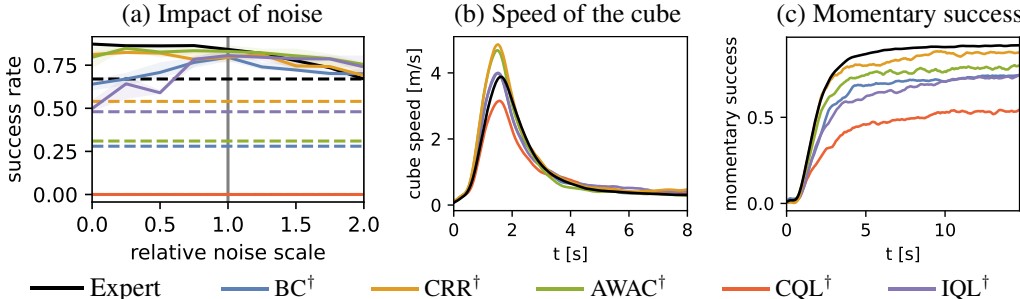

Figure 6: **Analysis of policy behavior.** (a) Success rate for simulated lifting after training on data with varying relative environment noise scale (1.0 corresponds to the noise level of the real system). The dashed lines indicate the performance on the real system. (b) Average speed of the cube over a pushing episode on the real system. Some offline RL algorithms learn to move the cube faster than the expert. (c) Momentary success during a pushing episode on the real system: the fraction of episodes during which the goal was achieved at a given point in time (trained on expert data).

are furthermore noisy and do not contain estimates of the velocity of the cube, training recurrent policies could potentially lead to an increase in performance.

To investigate the impact of noise on the performance of offline RL policies and the expert, we collected datasets in simulation for up to twice the noise amplitudes measured on the real system. Fig. 6 (a) shows that the performance of the expert and the offline RL policies degrades only slowly when increasing the noise scale, ruling out that noise is the sole explanation for the performance gap between simulated and real system. As delays in the observation and action execution are already implemented in the simulated environment, we conclude that other factors like more complex contact dynamics and elastic deformations of the fingertips and robot limbs are likely causing the larger performance gap between data and learned policies on the real robots.

To test how well the policies learned from the datasets generalize over instances of the robot hardware, we evaluated on a hold-out robot which was not used for data collection. We did not see a significant difference in performance (see Appendix A.3 for details on this experiment) suggesting that the datasets cover enough variations in the robot hardware to enable generalization to unseen robots.

In summary, CRR and AWAC generally perform best on the proposed datasets with IQL also being competitive after hyperparameter optimization. The performance gap between expert and offline RL is in general bigger on the real system, perhaps due to more challenging dynamics.

## 5 RELATED WORK

**Offline RL:** The goal of offline RL (Levine et al., 2020; Prudencio et al., 2022) is to learn effective policies without resorting to an online interaction by leveraging large and diverse datasets covering a sufficient amount of expert transitions. This approach is particularly interesting if interactions with the environment are either prohibitively costly or even dangerous.

Offline RL faces a fundamental challenge, known as the Distributional Shift (DS) problem, originating from two sources. There is a distribution mismatch in training, as we use the behavioral data for training, but the learned policy would create a different state-visitation distribution. The second problem is that during policy improvement, the learned policy requires an evaluation of the Q-function on unseen actions (out of distribution). An over-estimation of the Q-value on these out-of-distribution samples leads to learning of a suboptimal policy.

Several algorithmic schemes were proposed for addressing the DS problem: i) constraining the learned policy to be close to the behavior data (Fujimoto et al., 2019; Kumar et al., 2019b; Zhang et al., 2021; Kostrikov et al., 2021a); ii) enforcing conservative estimates of future rewards (Kumar et al., 2020; Yu et al., 2021; Cheng et al., 2022); and iii) model-based methods that estimate the uncertainty via ensembles (Janner et al., 2019; Kidambi et al., 2020).

Additionally, other approaches include: implicitly tackling the DS problem via (advantage-weighted) variants of behavioral cloning (Nair et al., 2020; Wang et al., 2020; Chen et al., 2020; Fujimoto & Gu, 2021) or even completely bypassing it either by removing the off-policy evaluation and

performing a constrained policy improvement using an on-policy Q estimate of the behavior policy (Brandfonbrener et al., 2021), or by approximating the policy improvement step implicitly by learning on-data state value function (Kostrikov et al., 2021b). An orthogonal research line considers combining importance sampling and off-policy techniques (Nachum et al., 2019b;a; Zhang et al., 2020; Xu et al., 2021), and recently (Chen et al., 2021; Janner et al., 2021) investigated learning an optimal trajectory distribution via transformer architectures.

The simplest strategy for learning from previously collected data is behavioral cloning (BC), which is fitting a policy to the data directly. Offline RL can outperform BC: i) on long-horizon tasks with mixed expert data, e.g., trajectories collected via an expert and a noisy-expert; and ii) with expert or near expert data, when there is a mismatch between the initial and the deployment state distribution.

**RL for dexterous manipulation:** Reinforcement learning was recently successfully applied to dexterous manipulation on real hardware (OpenAI et al., 2018; 2019; Allshire et al., 2022; Wang et al., 2022). These results rely on training with online RL in simulation, however, and are consequently limited by the fidelity of the simulator. While domain randomization (Tobin et al., 2017; Mandlekar et al., 2017; Peng et al., 2018) can account for a mismatch between simulation and reality in terms of physics parameters, it cannot compensate oversimplified dynamics. The challenges of real-world environments have been recognized and partly modeled in simulated environments (Dulac-Arnold et al., 2020). To overcome the sim-to-real gap entirely, however, data from real-world interactions is still required, in particular for robotics problems involving contacts.

**Offline RL datasets:** While offline RL datasets with data from simulated environments, like D4RL (Fu et al., 2020) and RL Unplugged (Gulcehre et al., 2020), have propelled the field forward, the lack of real-world robotics data has been recognized (Behnke, 2006; Bonsignorio & del Pobil, 2015; Calli et al., 2015; Amigoni et al., 2015; Murali et al., 2019). Complex contact dynamics, soft deformable fingertips and vibrations are particularly relevant for robotic manipulation but are not modeled sufficiently well in simulators used by the RL community (Todorov et al., 2012; Makoviychuk et al., 2021; Freeman et al., 2021). Recently, three small real-world datasets with human demonstrations for a robot arm with a gripper (using operational space control) have been proposed (Mandlekar et al., 2021). Two of them require only basic lifting and dropping while the third, more challenging task could not be solved with the available amount of data. For more challenging low-level physical manipulation, a dataset suitable for offline RL is still missing. We therefore provide the first real-robot dataset for dexterous manipulation which is sufficiently large for offline RL (one order of magnitude more data on real robots than prior work (Mandlekar et al., 2021)) and for which learned policies can easily be evaluated remotely on a real-robot platform.

**Affordable open-source platforms:** Our hardware platform is open source. Other affordable robotic open-source platforms are, for instance, a manipulator (Yang et al., 2019), a simple robotic hand and quadruped (Ahn et al., 2020). Since it is hard to set up and maintain such platforms, we provide access to our real platform upon request, and hope that this will bring the field forward.

**Remote benchmarks:** For mobile robotics, Pickem et al. (2017) propose the Robotarium, a remotely accessible swarm robotics research platform, and Kumar et al. (2019a) offer OffWorld gym consisting of two navigation tasks with a wheeled robot. Similarly, Duckietown (Paull et al., 2017) hosts the AI Driving Olympics (AI-DO-team, 2022).

## 6 CONCLUSION

We present benchmark datasets for robotic manipulation that are intended to help improving the state-of-the-art in offline reinforcement learning. To record datasets, we trained capable policies using online learning in simulation with domain randomization. Our analysis and evaluation on two tasks, Push and Lift, show that offline RL algorithms still leave room for improvement on data from real robotic platforms. We identified two factors that could translate into increased performance on our datasets: trajectory stitching and robustness to non-expert trajectories. Further, our analysis indicates that noise and delay alone cannot explain the larger gap between dataset and offline RL performance on real systems, underpinning the importance of real-robot benchmarks.

We invite the offline RL community to train their algorithms with the new datasets and test the empirical performance of the latest offline RL algorithms, e.g. (Kostrikov et al., 2021a; Xu et al., 2021; Kumar et al., 2021; Cheng et al., 2022), on real-robot hardware.

AUTHOR CONTRIBUTIONS

N.G., S.Bl., P.K., M.W., S.Ba., B.S., and G.M. conceived the idea, methods and experiments. G.M. initiated the project, M.W., S.Ba. and B.S. conceived the robotic platform. F.W. implemented the low-level robot control and parts of the submission system. N.G. trained the expert policies and created the datasets. N.G., S.Bl., and P.K. conducted the offline RL experiments, collected the results and analyzed them under the supervision of G.M. N.G. ran all experiments on the real systems. N.G. and F.W. wrote the software for downloading and accessing the datasets. N.G., S.Bl., P.K., F.W. and G.M. drafted the manuscript, and all authors revised it.

ACKNOWLEDGMENTS

We are grateful for the help of Thomas Steinbrenner in repairing and maintaining the robot cluster. Moreover, feedback by Arthur Allshire on training expert policies in simulation and by Huanbo Sun on domain randomization proved valuable. We acknowledge the support from the German Federal Ministry of Education and Research (BMBF) through the Tübingen AI Center (FKZ: 01IS18039B). Georg Martius is a member of the Machine Learning Cluster of Excellence, EXC number 2064/1 – Project number 390727645. Pavel Kolev was supported by the Cyber Valley Research Fund and the Volkswagen Stiftung (No 98 571). We thank the anonymous reviewers for comments which helped improve the presentation of the paper.

REPRODUCIBILITY STATEMENT

We publish the datasets we propose as benchmarks (sections 3.3 and B) and provide access to the cluster of real TriFinger platforms (section 2) we used for data collection. Submissions to the cluster do not require any robotics experience and can be made in the form of a Python implementation of a RL policy (section B.5). A simulated version of the TriFinger platform (Joshi et al., 2020) and a low-cost hardware variant are furthermore publicly available as open source (Wüthrich et al., 2021). For offline RL training we moreover use open-source software (Seno & Imai, 2021). Finally, we describe our hyperparameter optimization in detail and provide the resulting hyperparameters in section C.3.

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

# A    ADDITIONAL RESULTS

This section contains additional experimental results. In particular values for the average returns achieved by the algorithms are reported. These were omitted from the main text as the success rates are easier to interpret.

## A.1    RETURNS FOR THE PUSH TASK

We report the returns achieved on the Push task in Table S1. Note that for the simulated datasets the best performing algorithm in terms of return is not necessarily the same as the best performing one in terms of success rates (see Table 1). For instance, AWAC has the highest average return on Sim-Expert while CRR achieves a higher success rate. This has two reasons. First, the success is measured only at the final step of the episode, while the return is computed as the cumulative reward over time. Second, the return is dense as it is computed from the distance between cube and target, while the success rate is a sparse signal.

Table S1: **Push: Returns on the TriFinger-Push Datasets.** 'data' denotes the mean over the dataset. Average and standard deviation over five training seeds.

| Push-Datasets | data | BC | CRR | AWAC | CQL | IQL |
|---|---|---|---|---|---|---|
| Sim-Expert | 674 | $585 \pm 19$ | $636 \pm 20$ | $\mathbf{657 \pm 14}$ | $184 \pm 23$ | $631 \pm 18$ |
| Sim-Half-Expert | 674 | $535 \pm 18$ | $576 \pm 17$ | $\mathbf{586 \pm 14}$ | $226 \pm 12$ | $565 \pm 22$ |
| Sim-Weak&Expert | 512 | $460 \pm 50$ | $\mathbf{613 \pm 17}$ | $603 \pm 23$ | $311 \pm 29$ | $543 \pm 75$ |
| Sim-Mixed | 583 | $460 \pm 26$ | $205 \pm 100$ | $\mathbf{636 \pm 20^*}$ | $138 \pm 11$ | $588 \pm 25$ |
| Real-Expert | 660 | $563 \pm 34$ | $\mathbf{638 \pm 16}$ | $624 \pm 17$ | $514 \pm 37$ | $592 \pm 29$ |
| Real-Half-Expert | 660 | $546 \pm 29$ | $\mathbf{627 \pm 21}$ | $587 \pm 46$ | $471 \pm 26$ | $573 \pm 23$ |
| Real-Weak&Expert | 429 | $387 \pm 45$ | $\mathbf{622 \pm 41^*}$ | $568 \pm 29$ | $346 \pm 68$ | $555 \pm 16$ |
| Real-Mixed | 419 | $335 \pm 23$ | $373 \pm 41$ | $569 \pm 24$ | $206 \pm 18$ | $\mathbf{600 \pm 30}$ |

## A.2    RETURNS FOR THE LIFT TASK

In this section, we report the returns achieved on the Lift task (Table S2). Interestingly, although IQL achieves a significantly higher return on the Real-Weak&Expert dataset than CRR, its final success rate is lower. An analysis of the rollout statistics reveals that IQL often moves the cube close to the goal pose but not close enough to satisfy the success criteria (defined in section 3). CRR, on the other hand, on average deviates further from the goal pose than IQL but has a bigger fraction of rollouts in which it satisfies the success criteria. In summary IQL does better on average on this dataset but lacks precision in matching the goal pose.

## A.3    EVALUATION ON THE HOLD-OUT ROBOT

To quantify how well the policies obtained with offline RL generalize to unseen hardware, we evaluate them on a holdout robot which was not used for data collection. Table S3 shows the results for TriFinger-Push-Real-Expert and TriFinger-Lift-Real-Expert. The performance of the algorithms on the hold-out robot is within the performance margin of the other robots, suggesting that there is no significant difference between the different robots.

## A.4    IMPACT OF NOISE ON PERFORMANCE

As mentioned in section 4.1, we studied the impact of noise on the performance of the expert and the considered offline RL algorithms by recording a sequence of datasets with increasing noise scale in simulation. A relative noise scale of 1 corresponds to the variance measured on the actions and observations of the real system. Fig. S1 shows the success rate and return as a function of the noise scale up to a value of 2. As the performance of the expert and the offline RL policies degrades only slowly when increasing noise, we conclude that the stochasticity of the real system cannot be the only reason for the performance gap between the simulated and the real system. As delays in

Table S2: **Lift: Returns on the TriFinger-Lift Datasets.** 'data' denotes the mean over the dataset. Average and standard deviation over five training seeds.

| Lift-Datasets | data | BC | CRR | AWAC | CQL | IQL |
|---|---|---|---|---|---|---|
| Sim-Expert | 1334 | $1129 \pm 56$ | $1246 \pm 10$ | $\mathbf{1280 \pm 20}^*$ | $163 \pm 14$ | $1133 \pm 41$ |
| Sim-Half-Expert | 1337 | $1112 \pm 39$ | $\mathbf{1231 \pm 9}$ | $1211 \pm 19$ | $154 \pm 13$ | $744 \pm 40$ |
| Sim-Weak&Expert | 1133 | $791 \pm 43$ | $727 \pm 447$ | $\mathbf{1103 \pm 62}$ | $164 \pm 15$ | $943 \pm 56$ |
| Sim-Mixed | 1173 | $409 \pm 9$ | $604 \pm 79$ | $\mathbf{931 \pm 75}^{***}$ | $161 \pm 8$ | $572 \pm 24$ |
| Sim-Expert$^\dagger$ | 1334 | $1274 \pm 17$ | $1245 \pm 26$ | $\mathbf{1319 \pm 11}^*$ | $399 \pm 56$ | $1286 \pm 20$ |
| Sim-Half-Expert$^\dagger$ | 1337 | $1191 \pm 33$ | $1153 \pm 50$ | $\mathbf{1303 \pm 14}^*$ | $439 \pm 26$ | $1229 \pm 45$ |
| Sim-Weak&Expert$^\dagger$ | 1133 | $1040 \pm 22$ | $1087 \pm 49$ | $1120 \pm 23$ | $467 \pm 34$ | $\mathbf{1172 \pm 34}^*$ |
| Sim-Mixed$^\dagger$ | 1173 | $411 \pm 35$ | $593 \pm 93$ | $\mathbf{862 \pm 53}^{**}$ | $151 \pm 9$ | $564 \pm 31$ |
| Real-Smooth-Expert | 1206 | $915 \pm 36$ | $\mathbf{1059 \pm 54}$ | $1031 \pm 42$ | $143 \pm 33$ | $1002 \pm 19$ |
| Real-Expert | 1064 | $711 \pm 92$ | $\mathbf{1014 \pm 62}^*$ | $820 \pm 50$ | $283 \pm 38$ | $901 \pm 43$ |
| Real-Half-Expert | 1064 | $553 \pm 77$ | $\mathbf{837 \pm 58}$ | $613 \pm 107$ | $200 \pm 7$ | $759 \pm 101$ |
| Real-Weak&Expert | 851 | $346 \pm 21$ | $633 \pm 59$ | $397 \pm 71$ | $298 \pm 16$ | $\mathbf{827 \pm 105}^*$ |
| Real-Mixed | 862 | $272 \pm 56$ | $\mathbf{631 \pm 62}$ | $346 \pm 64$ | $207 \pm 5$ | $608 \pm 30$ |
| Real-Expert$^\dagger$ | 1064 | $676 \pm 47$ | $889 \pm 39$ | $747 \pm 44$ | $289 \pm 33$ | $\mathbf{899 \pm 40}$ |
| Real-Half-Expert$^\dagger$ | 1064 | $702 \pm 83$ | $813 \pm 57$ | $797 \pm 54$ | $312 \pm 65$ | $\mathbf{855 \pm 51}$ |
| Real-Weak&Expert$^\dagger$ | 851 | $437 \pm 47$ | $\mathbf{707 \pm 20}^*$ | $481 \pm 54$ | $269 \pm 33$ | $574 \pm 67$ |
| Real-Mixed$^\dagger$ | 862 | $288 \pm 73$ | $\mathbf{634 \pm 36}$ | $361 \pm 81$ | $192 \pm 28$ | $596 \pm 33$ |

Table S3: **Evaluation on hold-out robot.** Success rate on the Real-Expert datasets.

| Dataset | BC | CRR | AWAC | CQL | IQL |
|---|---|---|---|---|---|
| Push-Real-Expert | $0.80 \pm 0.04$ | $\mathbf{0.91 \pm 0.08}$ | $0.84 \pm 0.06$ | $0.61 \pm 0.05$ | $0.83 \pm 0.09$ |
| Lift-Real-Expert | $0.29 \pm 0.04$ | $\mathbf{0.64 \pm 0.05}^{**}$ | $0.31 \pm 0.08$ | $0.00 \pm 0.00$ | $0.24 \pm 0.07$ |

the observation and action execution are already implemented in the simulated environment, we hypothesize that other factors like more complex contact dynamics and elastic deformations of the fingertips and robot limbs are likely causing the larger performance gap between data and learned policies on the real robots.

AWAC and CRR perform consistently over a wide range of noise scales with a slight decrease in performance for high relative noise scales (probably due to a large variance of the estimated object pose). BC and IQL seem to struggle with the narrow data distribution generated by a deterministic environment but improve with increasing stochasticity. While the performance of BC drops significantly again for large noise scales, IQL becomes competitive in this regime.

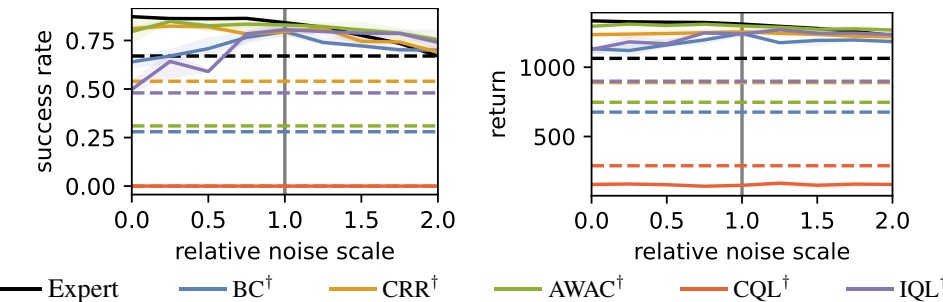

Figure S1: Success rate and return for simulated lifting for varying relative environment noise scales (1.0 corresponds to the noise level of the real system) when the algorithms were trained on data recorded with that noise scale using the expert policy. The dashed lines indicate the performance on the real system.

## A.5 BAR PLOTS

Fig. S2 and Fig. S3 show bar plots that summarize the performance of the algorithms on the two tasks in the simulated and real environment. Before averaging results from different datasets, we *normalized* the algorithm performance by the dataset performance, i.e., reaching the average success rate or return of the dataset corresponds to a value of 1.

While AWAC (and IQL on the real data) can exceed the behavior policy on average on the Push task, all algorithms fall short of matching the dataset performance on the challenging Lift datasets. While success rates on the Lift-Real datasets are particularly low, the returns indicate that CRR and IQL significantly outperform BC.

Since the hyperparameter optimization was done on the Lift-Sim-Weak&Expert dataset, it has the biggest impact on the performance on the Lift-Sim datasets. IQL in particular improves considerably there. The increase in performance on the Lift-Real datasets is considerably smaller, however. This suggests that optimizing the hyperparameters offline RL algorithms in simulation may have limited benefits on real environments.

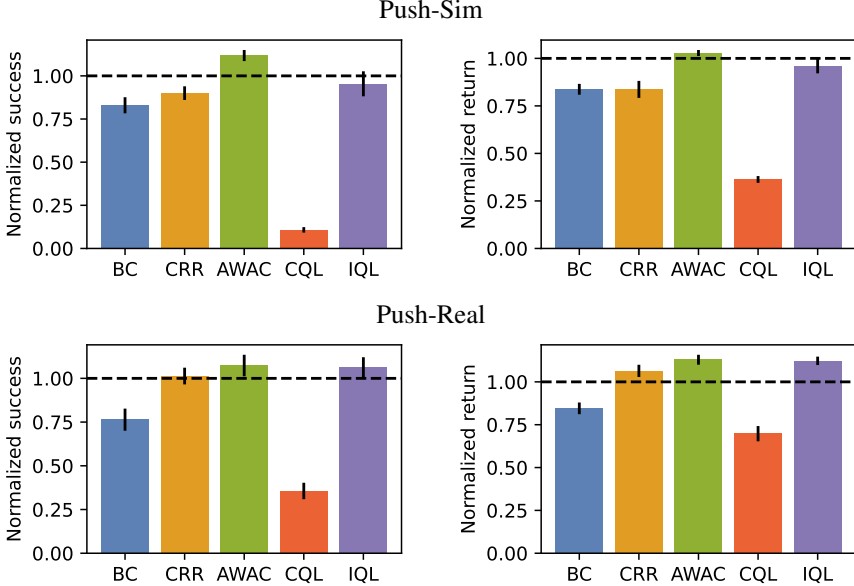

Figure S2: **Normalized performance for the Push task:** Success rates at the end of episodes (or returns) were normalized to the dataset success rate (or mean return) and then averaged over all datasets corresponding to a task (treating simulated and real data separately). Default hyperparameters were used for the Push task.

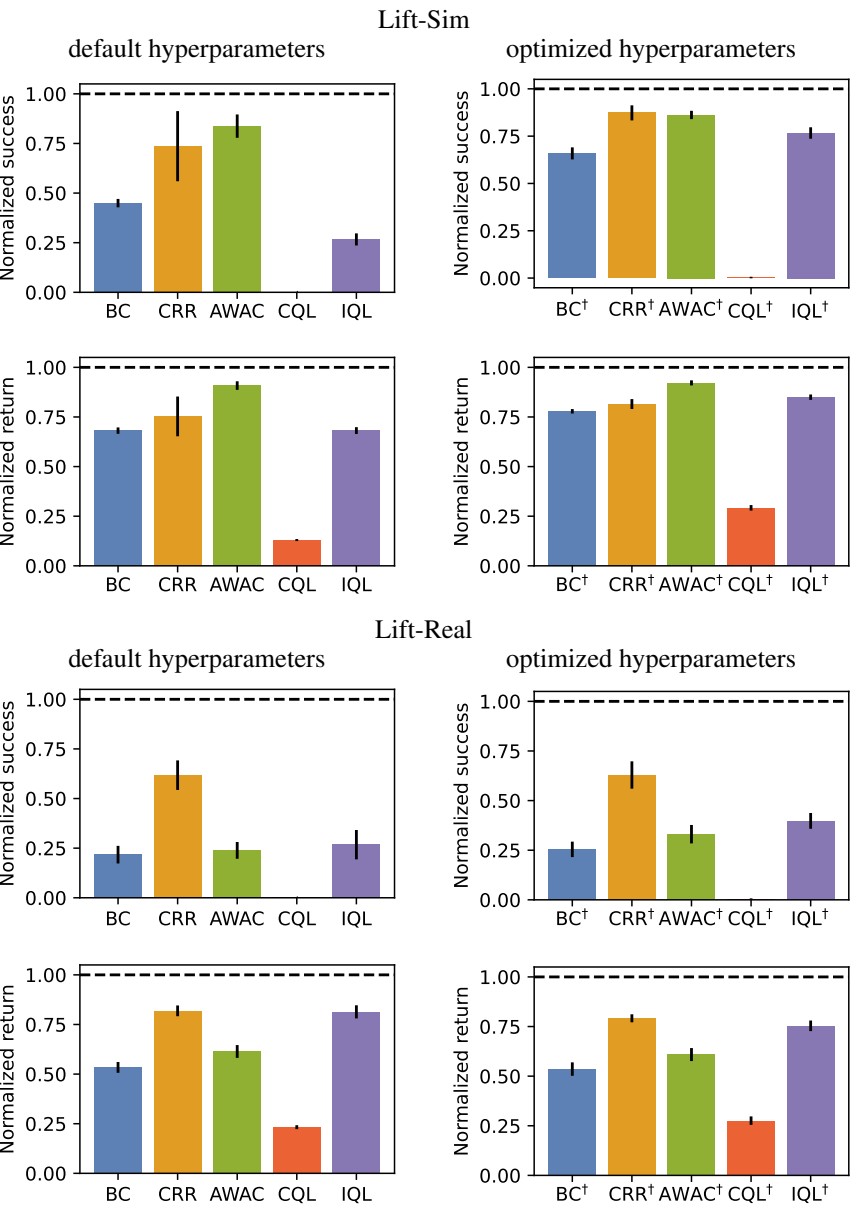

Figure S3: **Normalized performance for the Lift task:** Success rates at the end of episodes (or returns) were normalized to the dataset success rate (or mean return) and then averaged over all datasets corresponding to a task (treating simulated and real data separately).

## A.6 LEARNING CURVES

In this section, we provide learning curves for the offline RL algorithms on all datasets. Since the evaluation of all training checkpoints on the real robots is prohibitively expensive and time-consuming, we evaluate the checkpoints learned on the **real data** in the **simulated environment**. This gives an over-optimistic estimate of the learning performance on the real robots.

For the challenging Lift-Real-Weak&Expert task we performed a grid search for each algorithm (BC, CRR, AWAC, CQL, IQL) and selected the best hyperparameters. We present the grid search in Table S7, the corresponding optimal hyperparameters in Table S8, and the default hyperparameters in Table S9. We note that due to the poor performance of CQL, we expanded our grid search specifically for this algorithm (on the Push-Sim-Expert data) and selected the corresponding optimal hyperparameters as its default parameters (see Figure S16). Our newly performed gridsearch, as mentioned above, was unfortunately not leading to improvements for CQL. Similar difficulties are reported in Kumar et al. (2021) and are tackled via case distinction and appropriate regularization techniques. It would be interesting to test the preceding two algorithmic techniques on our datasets, once their implementation is integrated in the D3RLPY library (Seno & Imai, 2021).

We proceed by presenting the learning curves for the offline RL algorithms on the datasets: for the Lift task in section A.6.1 and for the Push task in section A.6.2. Shaded areas indicate the interval between the 0.25 and 0.75 quantiles.

### A.6.1  LIFT DATASETS

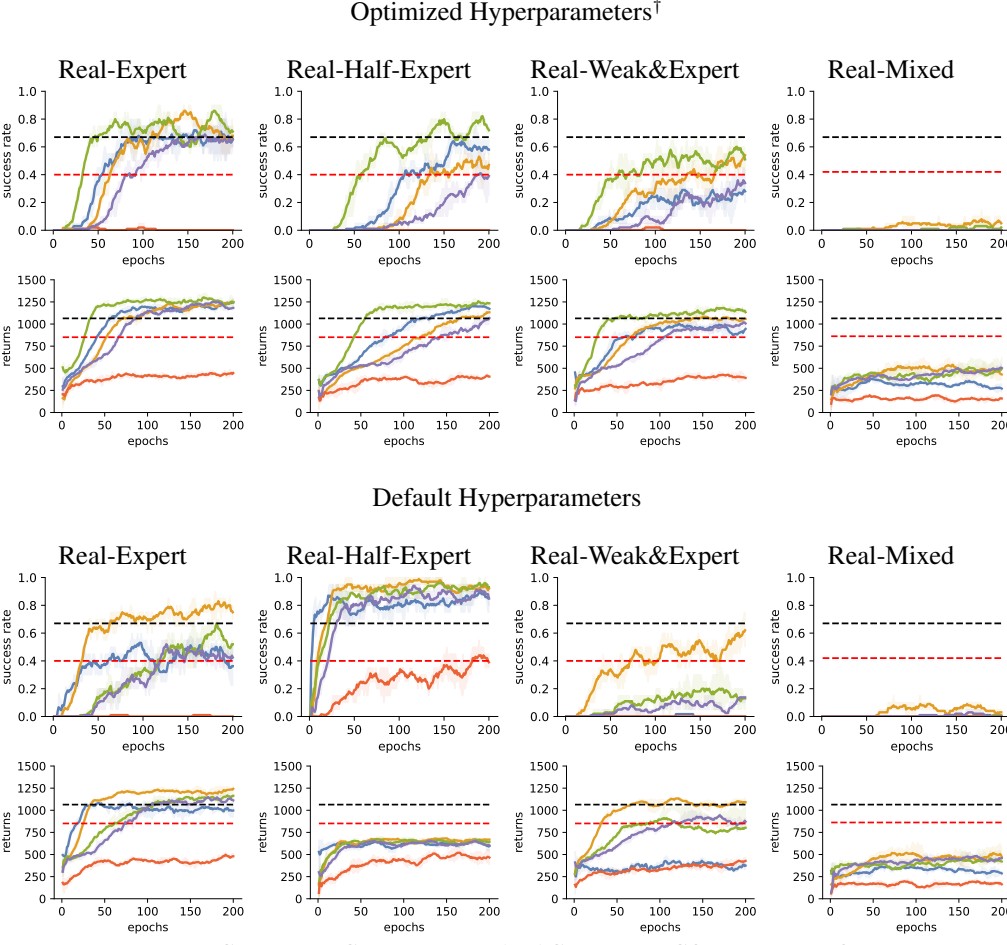

Figure S4: **Training curves for the Lift-Real datasets:** Offline RL algorithms are trained on the **real** datasets and are evaluated in the **simulated** environment. Success rates and returns for the optimized hyperparameters (in Table S8) and the default hyperparameters (in Table S9). *The selected hyperparameters, by the grid-search procedure (in Table S7), optimize the algorithm's **final average return** on Sim-Weak&Expert.* The black dashed line shows the dataset performance of Real-Expert and the red dashed line the performance of the Real-Weak&Expert (first 3 columns) and Real-Mixed (last column).

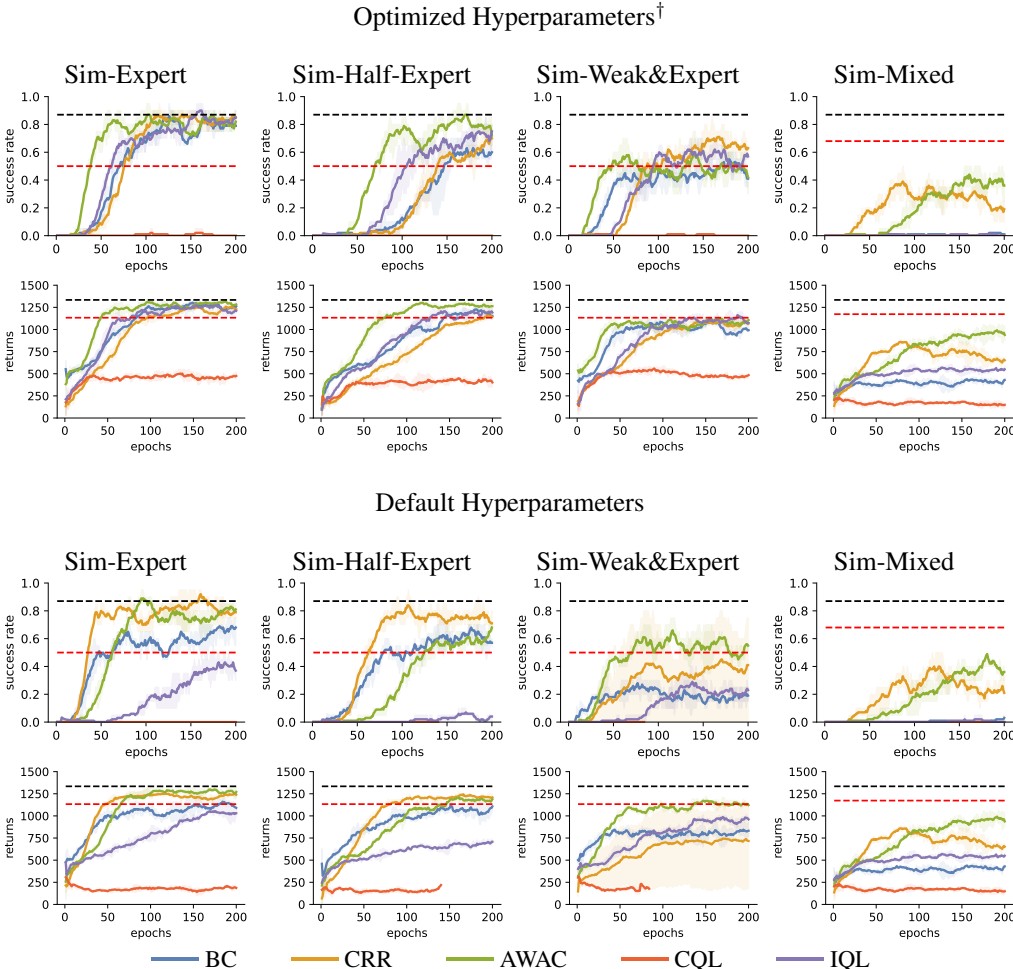

Figure S5: **Training curves for the Lift-Sim datasets:** Offline RL algorithms are trained on a **simulated** dataset and are evaluated in the **simulated** environment. Success rates and returns for the optimized hyperparameters (in Table S8) and the default hyperparameters (in Table S9). *The selected hyperparameters, by the grid-search procedure (in Table S7), optimize the algorithm's **final average return** on Sim-Weak&Expert*. The black dashed line shows the dataset performance of Sim-Expert and the red dashed line the performance of the Sim-Weak&Expert (first 3 columns) and Sim-Mixed (last column).

### A.6.2 PUSH DATASETS

Here, we use the **default** hyperparameters in Table S9, where for the CQL algorithm we selected hyperparameters optimized by a grid search (see Figure S16 for histograms). The *success rates* and *returns* are evaluated in the simulated environment.

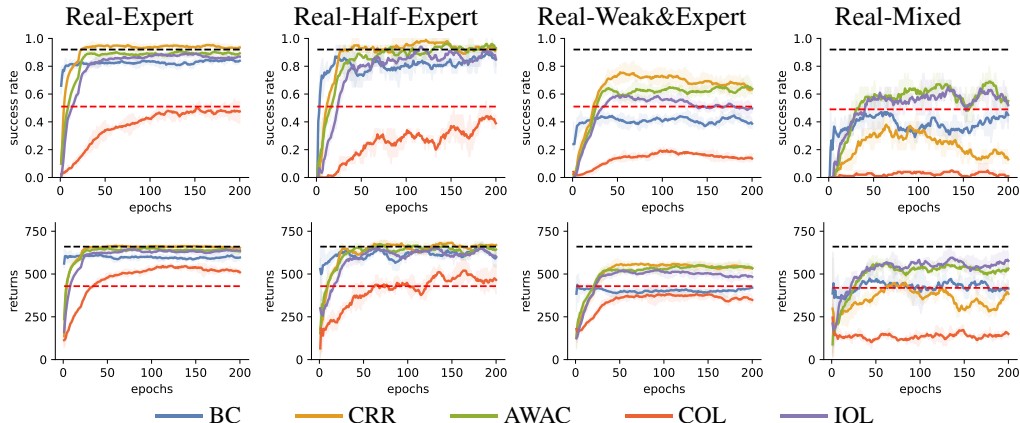

Figure S6: **Training curves for the Push-Real datasets:** Offline RL algorithms are trained on the **real** datasets and are evaluated in the **simulated** environment. The black dashed line shows the dataset performance of Real-Expert and the red dashed line the performance of the Real-Weak&Expert (first 3 columns) and Real-Mixed (last column).

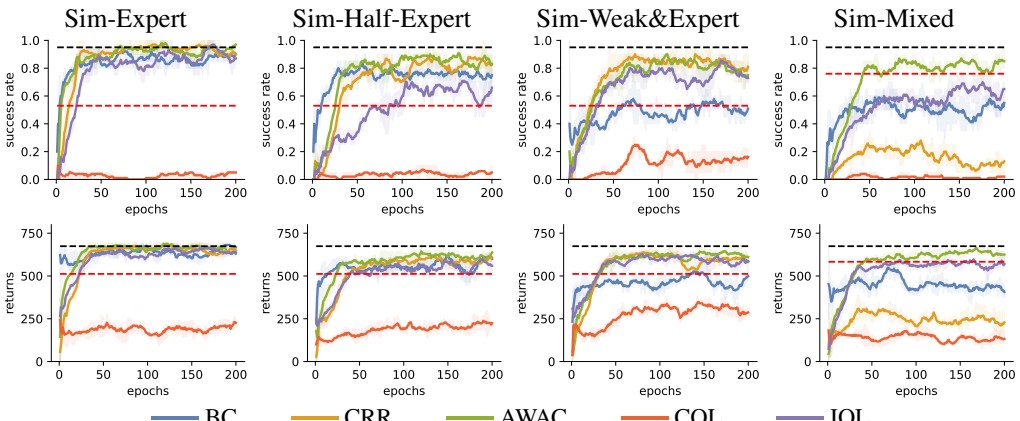

Figure S7: **Training curves for the Push-Sim datasets:** Offline RL algorithms are trained on a **simulated** dataset and are evaluated in the **simulated** environment. The black dashed line shows the dataset performance of Sim-Expert and the red the performance of the Sim-Weak&Expert (first 3 columns) and Sim-Mixed (last column).

## B  DATASETS

In this section we provide additional details on the datasets we collected. Section B.1 summarizes the data collection process, section B.2 contains a discussion of the different notions of success we report, section B.3 is dedicated to the reward and the terminals, and section B.4 provides a detailed analysis of the statistics of the datasets.

### B.1  DATA COLLECTION

Data is collected by running jobs on a cluster of TriFinger platforms without human intervention. Before the recording of each dataset the platforms were cleaned from particle dust to ensure consistent object tracking performance. Each job consists of the following steps:

1. Move fingers to initial state and sample new goal.
2. Run one episode/rollout with the selected policy and store transitions.
3. Move the cube away from the barrier with a pre-recorded trajectory of the fingers.
4. Repeat from 1. unless the desired number of episodes per job is reached.
5. Do a self-check including cameras, pose estimation and joints.
6. Approximately reset the cube to the center of the arena.

The number of episodes collected per job is eight for the Push task ($15\,\mathrm{s}$ per episode) and six for the Lift task ($30\,\mathrm{s}$ per episode). If a self-check fails, the corresponding robot is deactivated and the maintainers are alerted via mail. Finally, the data from all episodes is combined in a single HDF5 file following the conventions of D4RL (Fu et al., 2020).

The expert policies are obtained after the training in Isaac Gym (see section 3.2 and section C.1). The weak policies are training checkpoints at $210 \cdot 10^6$ training steps for pushing and $288 \cdot 10^6$ training steps for lifting. Gaussian noise is added to the actions with an amplitude of $0.2\,\mathrm{Nm}$ and an update frequency of 8 for pushing and $0.04\,\mathrm{Nm}$ with an update frequency of 4 for lifting.

The Mixed datasets are collected with policies from a range of training checkpoints where the total number of jobs was distributed as uniformly as possible over all training checkpoints. For pushing 22 checkpoints up to $668 \cdot 10^6$ training steps were used while for lifting 59 training checkpoints up to $1721 \cdot 10^6$ training steps were considered. Fig. S8 shows the return and success rate of these checkpoints on the real TriFinger cluster.

### B.2  THREE NOTIONS OF SUCCESS

We consider dexterous manipulation tasks which require the agent to match a goal position (for the Push task) or a goal pose (for the Lift task) with a tracked cube. The tolerance for goal achievement is $2\,\mathrm{cm}$ for the position and 22 degrees for the orientation. We define *momentary success* as achieving the goal at a single point in time, i.e., in an individual time step. This notion of success is rather weak, however, as achieving the goal pose for a short amount of time is significantly easier than maintaining it. For the pushing task in particular, it is much more likely to move through the goal by accident than to stabilize the cube after having reached it. When lifting the cube, it is quite challenging to maintain a stable grasp due to the variance of the object pose estimate which introduces a considerable amount of noise to the observations. We therefore define *success* as achieving the goal at the end of the episode which is only likely to happen when maintaining the goal pose for an extended period of time.

From the perspective of offline RL, however, it is highly relevant whether parts of the trajectories in the datasets lead to success, even if it is short-lived. As offline RL algorithms can, in principle, combine information from several trajectories, even trajectories that do not end with goal achievement may contain enough information to learn a successful policy. We therefore also report *transient success* which indicates whether the goal has been achieved at any time during the episode and *mean momentary success* which corresponds to the fraction of time steps during which the goal has been achieved.

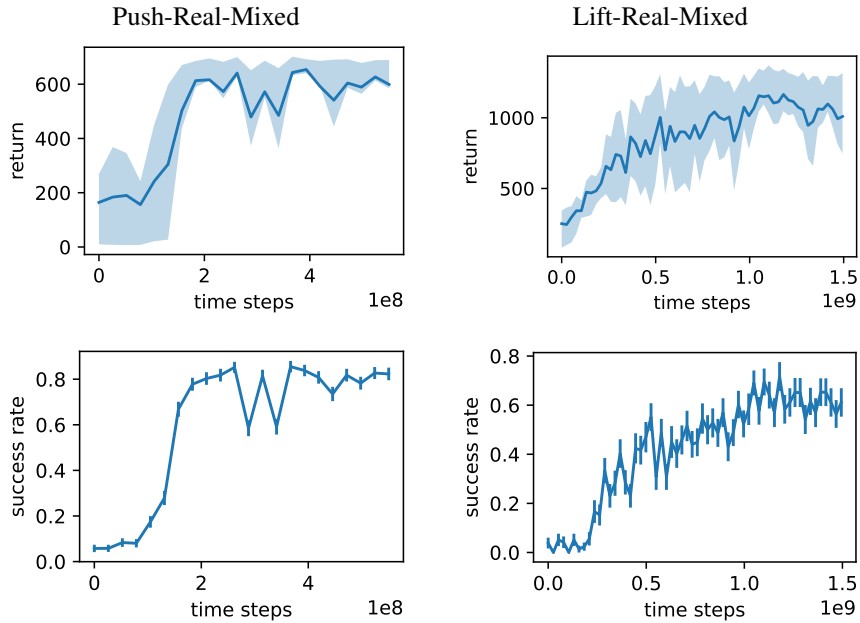

Figure S8: Return and success rates of the checkpoints used for collecting the Real-Mixed datasets. Note that the checkpoints were obtained by training in Isaac Gym (see section 3.2 and section C.1) and were evaluated on the **real** system. The shaded areas indicate the interval between the 0.25 and 0.75 quantiles of the return whereas the error bars represent the standard error of the mean of the success rate.

### B.3 REWARD AND TERMINALS

As already mentioned in section 3 of the main text, we use a logistic kernel Hwangbo et al. (2019); Allshire et al. (2022)

$$k(x) = \frac{b + 2}{\exp(a\|x\|) + b + \exp(-a\|x\|)} \tag{S1}$$

to define the reward where $\|\cdot\|$ denotes the Euclidean norm. For the Push task, we apply it to the difference between the achieved and the desired cube position. Since we also want to take the orientation of the cube into account for the Lift task, we apply $k$ separately to the differences between the desired and achieved corner points (or keypoints) of the cube and average over the results Allshire et al. (2022). We use $a = 30$ and $b = 2$. See Fig. S9 for a visualization of the reward function.

By default, the terminals in the dataset (a Boolean indicating whether a terminal state has been reached) are never set. We chose this default setting to avoid problems due to state aliasing: Episodes

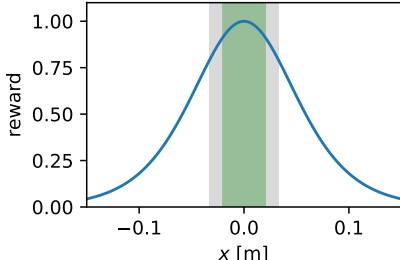

Figure S9: Reward as a function of the Euclidean distance between the desired and achieved position (for the Push task) or keypoint (for the Lift task). The shaded grey area corresponds to a cube centered at the goal and the green area corresponds to the goal achievement threshold (2 cm).

Table S4: Overview of the TriFinger offline RL datasets.

| task | dataset | overall duration [h] | #episodes | #transitions [$10^6$] | episode length [s] |
|------|---------|---------------------|-----------|----------------------|--------------------|
| Push- | Sim-Expert | 16 | 3840 | 2.8 | 15 |
| | Sim-Half-Expert | 8 | 1920 | 1.4 | 15 |
| | Sim-Weak&Expert | 16 | 3840 | 2.8 | 15 |
| | Sim-Mixed | 16 | 3840 | 2.8 | 15 |
| | Real-Expert | 16 | 3840 | 2.8 | 15 |
| | Real-Half-Expert | 8 | 1920 | 1.4 | 15 |
| | Real-Weak&Expert | 16 | 3840 | 2.8 | 15 |
| | Real-Mixed | 16 | 3840 | 2.8 | 15 |
| Lift- | Sim-Expert | 20 | 2400 | 3.6 | 30 |
| | Sim-Half-Expert | 10 | 1200 | 1.8 | 30 |
| | Sim-Weak&Expert | 20 | 2400 | 3.6 | 30 |
| | Sim-Mixed | 20 | 2400 | 3.6 | 30 |
| | Real-Smooth-Expert | 20 | 2400 | 3.6 | 30 |
| | Real-Expert | 20 | 2400 | 3.6 | 30 |
| | Real-Half-Expert | 10 | 1200 | 1.8 | 30 |
| | Real-Weak&Expert | 20 | 2400 | 3.6 | 30 |
| | Real-Mixed | 20 | 2400 | 3.6 | 30 |

last for a finite number of time steps $H$, and the RL objective is to maximize the discounted return (or cumulative reward)

$$J = \sum_{t=0}^{H-1} \gamma^t r_t \, , \qquad (S2)$$

where $\gamma$ denotes the discounting factor and $r_t$ the reward in step $t$. A Markov state therefore has to keep track of how much time remains until the episode ends. The observation, on the other hand, does not contain this information. This omission is a deliberate choice as we want to obtain a policy which is independent of the time step. This makes it impossible, however, to accurately estimate the value based on the observation if $\gamma$ is too large. Intuitively, without access to the time step $t$, the agent cannot know whether a cube far away from the goal corresponds to a large expected return to go because it is the beginning of an episode or a small expected return to go because there is no time left to move the cube and accumulate reward. See Pardo et al. (2018) for a more detailed discussion of issues arising from training with a finite horizon.

A practical solution to this problem is to not set the terminals and choose a gamma which is appropriate for the time scale of the task (for offline learning we chose $\gamma = 0.99$). This choice hides the episodic nature of the task from the agent which results in good performance while avoiding a dependence of the policy on time. It may, however, sacrifice optimality in some corner cases like dropping the cube close to the end of the episode when there is not enough time to flip the cube over before lifting it again. Nevertheless, as we provide a flag to set the terminals at the episode ends and as it is straight forward to augment the observation with the intra-episode time step, the datasets are also suitable for experiments with time-dependent policies.

### B.4 DATASET ANALYSIS

**Overview:** We provide an overview of the various dataset types and their properties in Table S4.

**Statistics:** Table S5 summarizes the statistics of the TriFinger datasets. While the success rate and the mean return constitute a limit to what can be achieved with pure imitation learning, the high numbers for the transient success rate (fraction of episodes in which the goal was achieved at least in one time step but not necessarily at the end of the episode) indicate that offline RL can potentially outperform the behavior policy significantly on these datasets.

**Impact of variations between robots:** Fig. S10 compares the success rates of the expert policies on the individual robots used for data collection. While the performance differences between the robot instances are significant, the expert policies perform reasonably well on all of them, achieving at least 80% success rate on the Push task and 60% on the Lift task on all robots.

Table S5: Statistics of the proposed TriFinger offline RL datasets: TriFinger-Push and TriFinger-Lift. Definitions of the success rates and the reward can be found in section B.2 and section B.3, respectively.

| TriFinger-Push | success rate | mean momentary success rate | transient success rate | mean return |
|---|---|---|---|---|
| Sim-Expert | 0.95 | 0.87 | 0.95 | 674 |
| Sim-Half-Expert | 0.94 | 0.86 | 0.94 | 667 |
| Sim-Weak&Expert | 0.53 | 0.48 | 0.86 | 512 |
| Sim-Mixed | 0.76 | 0.68 | 0.77 | 583 |
| Real-Expert | 0.92 | 0.78 | 0.98 | 660 |
| Real-Half-Expert | 0.92 | 0.78 | 0.98 | 660 |
| Real-Weak&Expert | 0.51 | 0.43 | 0.72 | 429 |
| Real-Mixed | 0.49 | 0.40 | 0.63 | 419 |

| TriFinger-Lift | success rate | mean momentary success rate | transient success rate | mean return |
|---|---|---|---|---|
| Sim-Expert | 0.87 | 0.77 | 0.97 | 1334 |
| Sim-Half-Expert | 0.88 | 0.78 | 0.98 | 1337 |
| Sim-Weak&Expert | 0.50 | 0.44 | 0.93 | 1133 |
| Sim-Mixed | 0.68 | 0.60 | 0.84 | 1173 |
| Real-Smooth-Expert | 0.64 | 0.53 | 0.82 | 1206 |
| Real-Expert | 0.67 | 0.52 | 0.87 | 1064 |
| Real-Half-Expert | 0.68 | 0.52 | 0.86 | 1064 |
| Real-Weak&Expert | 0.40 | 0.30 | 0.66 | 851 |
| Real-Mixed | 0.42 | 0.32 | 0.65 | 862 |

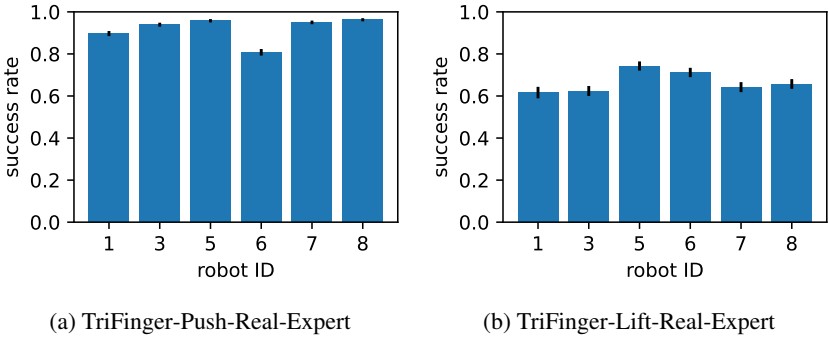

(a) TriFinger-Push-Real-Expert      (b) TriFinger-Lift-Real-Expert

Figure S10: Success rates on the individual robots. The transitions were recorded with policies trained with PPO in simulation. While there are significant differences between the success rates, the expert policies achieve at least 80% and 60% on every robot for TriFinger-Push-Real-Expert and TriFinger-Lift-Real-Expert, respectively.

**Distribution of returns:** Fig. S11 shows the distribution of the episode returns for the datasets collected on the real system. It reveals that the expert policy on the Push task performed more consistently on the real robot than its counterpart for the Lift task. Qualitatively, this can be attributed to either (i) not being able to flip the cube to the approximately correct orientation, (ii) failing to establish a stable grasp on the cube, or (iii) dropping it after already having lifted it. Flipping the cube on the ground might fail due to incomplete modeling of interactions between the fingertips and the cube in the rigid body physics simulator or because it is sensitive to the value of sliding friction, the noise on the object pose estimate makes it difficult to maintain a stable grasp after having lifted the object.

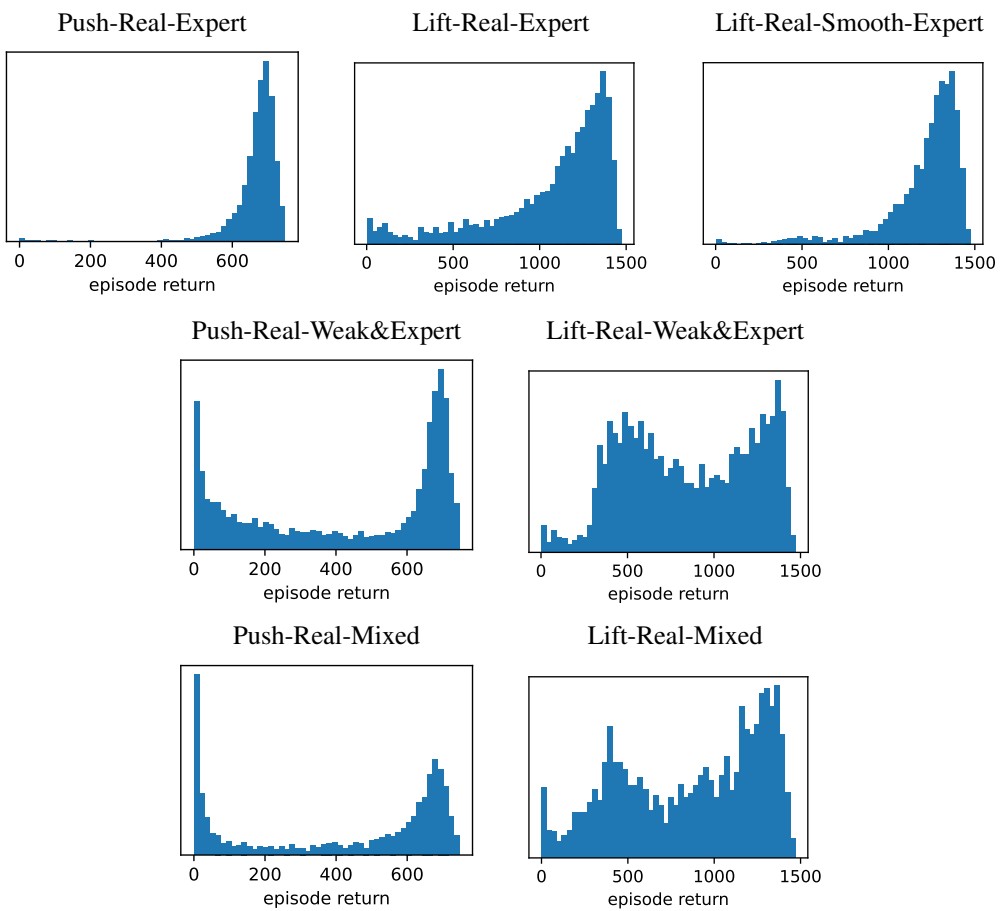

Figure S11: Distribution of the episode returns in the datasets recorded on the real system.

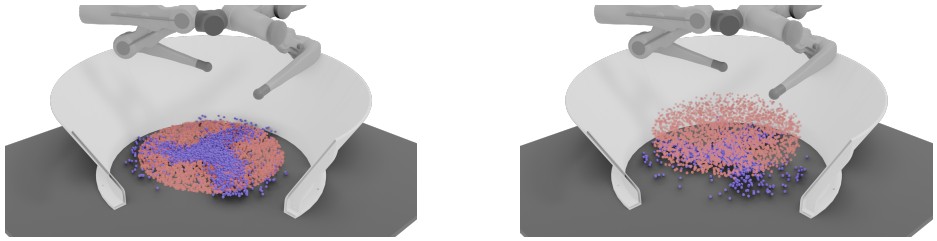

Figure S12: **Dataset visualization**. Initial cube positions (blue) and goals (red) in the real datasets for the Push task (left) and Lift task (right). Note that we avoid initial positions close to the barrier.

**Initial cube position and goal distributions:** Fig. S12 visualizes the distribution of the initial cube position (blue) and the goal position (red) for the Push and Lift tasks. The goal positions are sampled on the ground for the Push task and in the air for the Lift task. The distribution of the initial cube position results from the reset procedure which removes the cube from the boundaries by moving the fingers along a pre-recorded trajectory.

**Action statistics:** Fig. S13 shows the distribution of actions of the expert policy that recorded TriFinger-Push-Real-Expert and of a policy learned from this data by AWAC.

**Reset procedure:** As mentioned in section 3.3 of the main text, we removed the cube from the barrier between episodes because we observed that the expert policies we trained in simulation struggled with retrieving the cube from the barrier. Fig. S14a shows a histogram of the distance between the center of the cube and the arena center. While the reset procedure is sufficiently stochastic

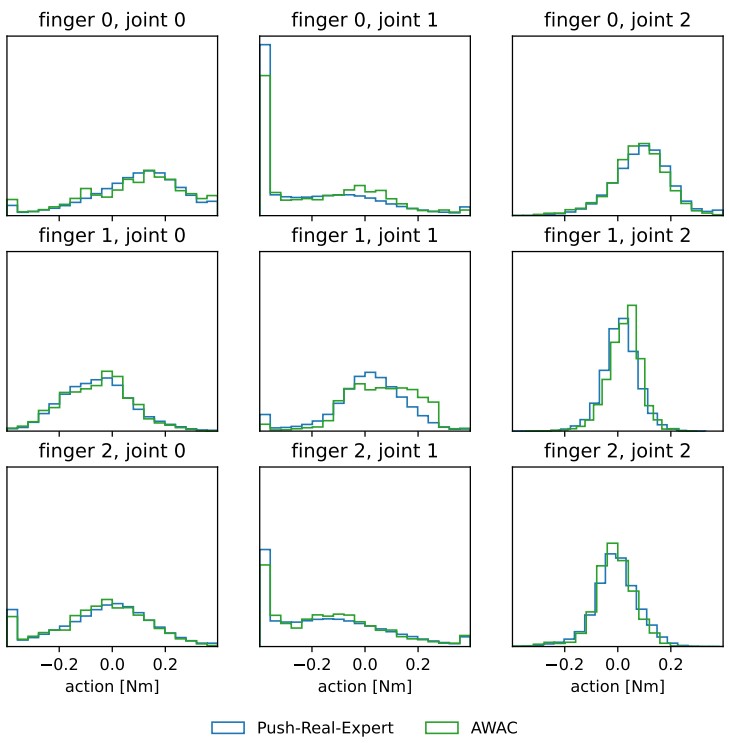

Figure S13: Distribution of action components, i.e., desired torques, for the expert policy that recorded the TriFinger-Push-Real-Expert dataset and for a policy learned with AWAC from this dataset. Averaged over five offline RL seeds.

to randomize the initial cube position (see Fig. S12), it clearly removes the cube from the barrier in the majority of all resets. More precisely, only in 3% of all resets the cube center was within 6 cm of the boundary (the cube has a width of 6.5 cm).

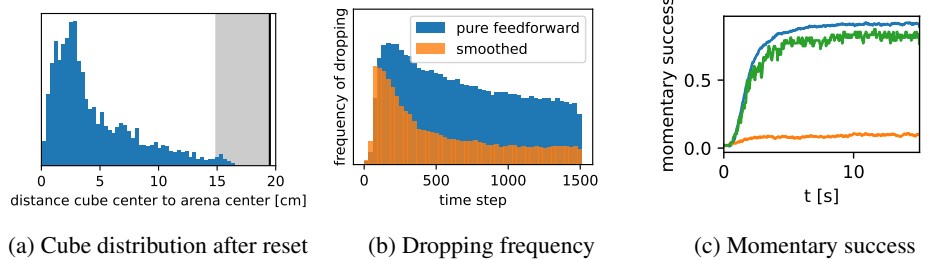

(a) Cube distribution after reset     (b) Dropping frequency     (c) Momentary success

Figure S14: (a) Histogram of the distance between the center of the cube and the arena center after the reset for the Lift-Real-Expert dataset. In the shaded gray area the cube can potentially touch the barrier which begins at the vertical black line. (b) Dropping frequency over a real lift episode. 'Pure feedforward' refers to directly applying the output of an MLP policy whereas 'smoothed' denotes a policy with an Exponential Moving Average applied. (c) Comparison of the momentary success rate of a policy learned from Push-Real-Expert (AWAC in green) with the expert policy (PPO-Expert in blue) that recorded the data. The momentary success rate captures how often the policy achieved the goal at a time step averaged over all episodes. The momentary success rate of the weak policy used in Push-Real-Weak&Expert is shown in orange.

**Dropping frequency – pure feedforward vs. smoothed:** One reason for the lower success rate on the Lift task as compared to the Push task is frequent dropping of the cube. This can be the result of vibrations of the fingers which can occur when applying policies trained in simulation on the

real robots or of shaking caused by the noisy pose estimate for the cube. We found that applying a low-pass filter, more precisely an Exponential Moving Average, helps with both problems. Fig. S14b shows the frequency at which the cube is dropped over the course of a lifting episode. Dropping the cube is defined as reaching the height of the goal pose up to a tolerance of $2\,\mathrm{cm}$ at some point during the episode and then dropping the cube flat on the ground (up to a tolerance of $1\,\mathrm{cm}$). Smoothing the actions before applying them clearly helps with avoiding dropping, in particular later in the episode. Note, however, that the smoothing has to be applied already during training in simulation for the policy to adapt to it. Despite the action smoothing the dropping rate does not reach zero. This can be partly attributed to unstable grasps that lead to slipping on the real robot.

### B.5 USING THE DATASETS

We provide an easy-to-use Python package that is compatible with a popular collection of offline RL datasets Fu et al. (2020). To use the datasets, it is sufficient to clone the repository, instantiate the desired environment and call a method that returns the dataset. The correct dataset is then downloaded automatically. We also provide options for filtering the observations and for converting them to flat arrays.

The following example code demonstrates how (parts of) a dataset can be loaded:

```python
import gymnasium as gym

import numpy as np

import trifinger_rl_datasets

env = gym.make(
    "trifinger-cube-push-real-expert-v0",
    disable_env_checker=True,
    visualization=False,
)

# load data at timesteps specified in array
dataset_part = env.get_dataset(indices=np.array([42, 1000, 5000]))

# load data corresponding to a range of timesteps
dataset_part = env.get_dataset(rng=(1000, 2000))

# load the whole dataset
dataset = env.get_dataset()
```

For installation instructions and further details we refer to the repository of the Python package at `https://github.com/rr-learning/trifinger_rl_datasets`.

### B.6 SUBMITTING A POLICY TO THE ROBOT CLUSTER

Access to the robot cluster can be requested at `https://webdav.tuebingen.mpg.de/trifinger/`. The policy has to be implemented in a GitHub repository following a fixed interface. We recommend adapting the example package available at `https://github.com/rr-learning/trifinger-rl-example`.

Table S6: Performance of the expert policies in the Isaac Gym environment. The numbers for the success rate and return are not directly comparable to those in the PyBullet simulator and on the real robot for two reasons: (i) In Isaac Gym Lift episodes last for $15\,\mathrm{s}$ instead of $30\,\mathrm{s}$, and (ii) the return in the Isaac Gym environment is contains auxiliary reward terms as described in Allshire et al. (2022) and section 3.2.

|      | success rate | return | #training steps |
|------|:---:|:---:|:---:|
| Push | 0.99 | 629 | $0.83 \cdot 10^9$ |
| Lift | 0.87 | 589 | $1.72 \cdot 10^9$ |

## C  TRAINING

### C.1  TRAINING EXPERT POLICIES IN SIMULATION

In addition to the modifications mentioned in section 3.2, we also ported training from Isaac Gym 2 to the current version Isaac Gym 3 (Makoviychuk et al., 2021). We furthermore increased the torque range from $[-0.36\,\mathrm{Nm}, 0.36\,\mathrm{Nm}]$ to $[-0.397\,\mathrm{Nm}, 0.397\,\mathrm{Nm}]$ to make sure the fingers are strong enough to lift the cube when supporting it from below. To make the training environment resemble the data collection setting on the real robots, we furthermore implemented an adjustable delay between when an observation is received and when the action based on this observation is applied for the first time. The success rates and returns reached after training are shown in Table S6.

Note that the success rates we give for the lifting task on the real robot are not directly comparable to the ones reported in Allshire et al. (2022) as our lifting task is more challenging. While Allshire et al. (2022) evaluate success after $60\,\mathrm{s}$[3], we evaluate after $30\,\mathrm{s}$. As policies usually need several attempts to flip the cube to roughly the correct orientation and for picking it up, the success rate after $30\,\mathrm{s}$ is lower in general. We chose the shorter episode length because it is, in principle, sufficient to solve the task and because we wanted to avoid episodes which consist, to a large part, of the cube being held in place. Moreover, unlike Allshire et al. (2022), we do not push the cube to the center of the arena before each episode but only remove it from the barrier.

### C.2  TRAINING WITH OFFLINE RL ON THE DATASETS

We use the implementations of BC, CRR, AWAC, CQL and IQL provided by the open-source library D3RLPY (Seno & Imai, 2021). The code is available at `https://github.com/takuseno/d3rlpy` and the documentation can be found at `https://d3rlpy.readthedocs.io/`. For our experiments we used versions 1.1.0 and 1.1.1 of D3RLPY. The used hyperparameters and the performed optimization are discussed in the next section.

### C.3  HYPERPARAMETERS

We performed a grid search over hyperparameters for all algorithms as documented in Table S7. The hyperparameter setting with the highest performance in terms of final average return on Lift-Sim-Weak&Expert was selected, as listed in Table S8. In the paper, the results with optimized parameters are marked with a [†]. Otherwise, the default parameters were used, as listed in Table S9.

The rest of the section contains a detailed analysis of the grid search.

---

[3]Personal communication

Table S7: **Hyperparameter grid search (on dataset Lift-Sim-Weak&Expert)**. Each algorithm is trained with the same two seeds (sampled uniformly at random).

| Algorithms | Parameters |
|---|---|
| AWAC | {(actor_learning_rate=R, critic_learning_rate=R) : R∈{1.5E-4, 3.0E-4, 6.0E-4}}; batch_size∈ {256, 512}; lam∈ {0.3, 1.0, 3.0} |
| BC | learning_rate={1.5E-4; 3.0E-4; 6.0E-4}; batch_size∈ {256, 512} |
| CRR | {(actor_learning_rate=R, critic_learning_rate=R) : R∈{1.5E-4, 3.0E-4, 6.0E-4}}; batch_size∈ {256, 512}; beta∈ {0.25, 1.0, 4.0} |
| CQL | {(actor_learning_rate=R, critic_learning_rate=3.0*R, initial_alpha=T[1], alpha_learning_rate={R if T[0] else 0}, temp_learning_rate=R, alpha_threshold=T[2]) : R∈{5.0E-5, 1.0E-4}, T∈{[true, 1.0, 1.0], [true, 1.0, 5.0], [true, 1.0, 10.0], [false, 0.3, 10.0], [false, 1.0, 10.0], [false, 3.0, 10.0]} }; conservative_weight∈{2.5, 5.0, 10.0, 20.0} |
| IQL | {(actor_learning_rate=R, critic_learning_rate=R) : R∈{1.5E-4, 3.0E-4, 6.0E-4}}; batch_size∈ {256, 512}; expectile∈ {0.7, 0.8, 0.9}; weight_temp∈ {3.0, 10.0} |

Table S8: **Optimized Hyperparameters using grid search** Table S7.

| Algorithms | Parameters |
|---|---|
| AWAC[†] | actor_learning = critic_learning_rate = 0.00015; batch_size=256; lam=3.0 |
| BC[†] | learning_rate=0.00015; batch_size=512 |
| CRR[†] | actor_learning = critic_learning_rate = 0.00015; batch_size=256; beta=1.0 |
| CQL[†] | actor_learning_rate=0.0001; critic_learning_rate=0.0003; initial_alpha=1.0; conservative_weight=20.0; alpha_learning_rate=0.0; action_scaler: minmax |
| IQL[†] | actor_learning_rate = critic_learning_rate= 0.00015; batch_size=256; expectile=0.9; weight_temp=3.0 |

Table S9: **Default hyperparameters**. All algorithms except BC (with batch_size=100 and without a critic) have batch_size=256 and n_critics=2. Note that due to bad performance we optimized CQL's parameters on Push-Sim-Expert with an extensive grid search as shown in Fig. S16. For all other algorithms, we used the default values of the implementation.

| Algorithms | Parameters |
|---|---|
| AWAC | actor_learning = critic_learning_rate = 0.0003; batch_size=1024; lam=1.0 |
| BC | learning_rate=0.001; batch_size=100 |
| CRR | actor_learning = critic_learning_rate = 0.0003; batch_size=256; beta=1.0 |
| CQL | actor_learning_rate=0.0001; critic_learning_rate=0.0003; initial_alpha=1.0; conservative_weight=20.0; alpha_learning_rate=0.0; action_scaler: minmax |
| IQL | actor_learning_rate = critic_learning_rate= 0.0003; batch_size=256; expectile=0.7; weight_temp=3.0 |

In Fig. S15 we present the returns and success rates for each of the hyperparameter settings for the Lift task in simulation. We see that the different algorithms have very different sensitivity to their hyperparameters. For CRR a large fraction of parameters leads to good results. For AWAC the sensitivity is a bit higher. IQL seems to degrade more gracefully with changed parameters. For CQL we were unable to find good hyperparameters, despite running 48 configurations. The performance of the individual runs over training time are shown in Fig. S17.

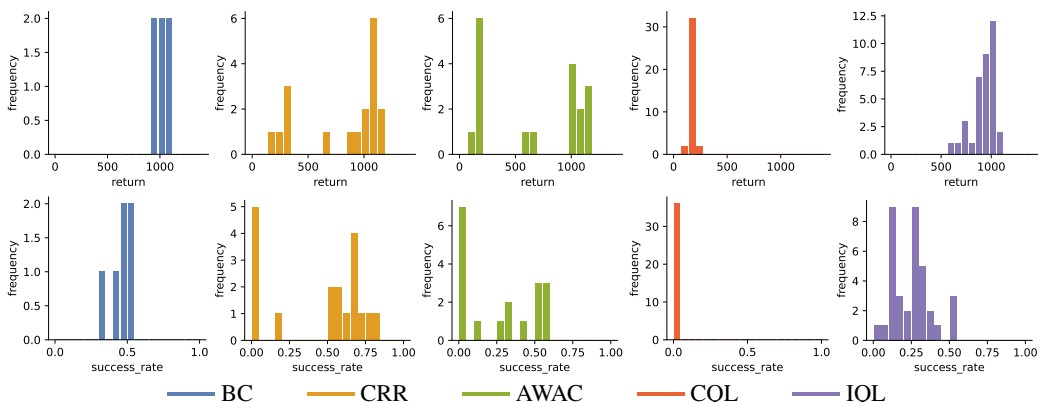

Figure S15: Hyperparameter grid search: Histogram of **returns** and **success rates** for the Lift-Sim-Weak&Expert task.

On the Push task, we ran an even larger hyperparameter scan with 405 configurations for CQL, as presented in Fig. S16. Even for this much simpler task, we see that the majority of parameters yield low success rates. In addition, we note that the training became quite unstable for alpha_lr > 0.0 and due to this, we conducted our main grid-search with alpha_lr = 0.0. Then, we chose the best hyperparameter configuration as the default setting for CQL.

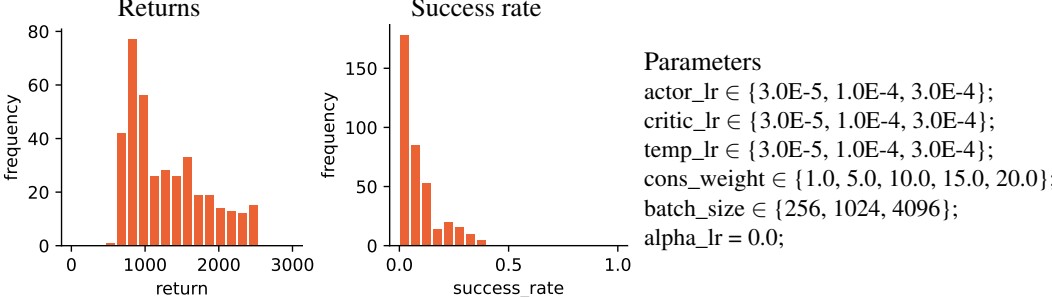

Figure S16: Hyperparameter grid search on the Push-Sim-Expert task for **CQL**. Shown is the histogram of returns and success rates for the 405 hyperparameter settings, as defined on the right.

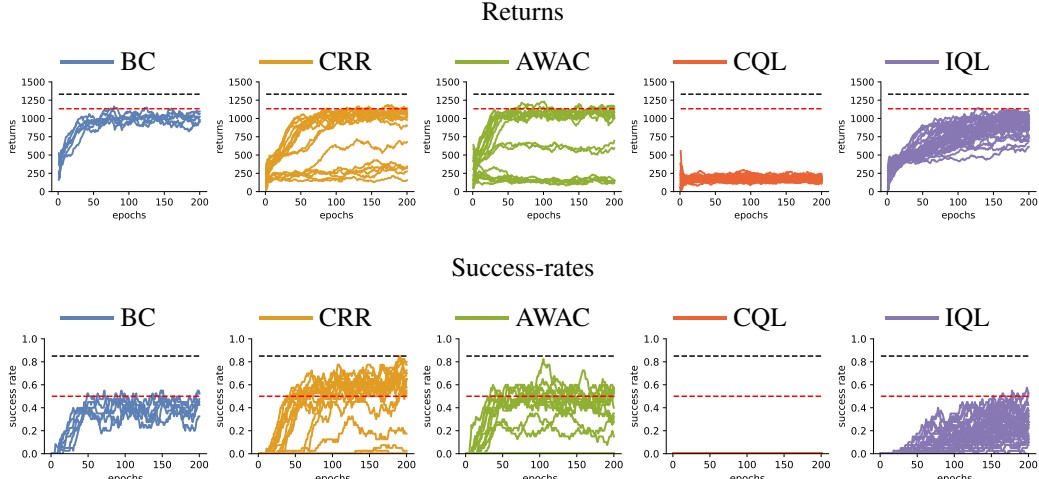

Figure S17: **Lift-Sim-Weak&Expert**. Hyperparameter grid search. **Returns** (top row) and **success rate** (bottom row). Every curve corresponds to one parameter configuration (averaged over 2 seeds). The black dashed line shows the dataset performance of Sim-Expert and the red dashed line the performance of the Sim-Weak&Expert.

## D   POLICY EVALUATION

### D.1   EVALUATION ON THE REAL SYSTEM

On the real platforms, we report numbers for 48 trials for the Push task and 36 trials for the Lift task, each for 5 training seeds. Each algorithm and seed is evaluated on 5 to 6 robots with the same set of 6 (for the Lift task) to 8 (for the Push task) goals per robot (computed from the robot ID). The success rate and return of each algorithm and seed is computed from the resulting trials. The mean and standard deviations of the success rates and returns of each algorithm is computed across seeds.

### D.2   EVALUATION IN THE SIMULATOR

In the simulated environment, we perform 100 testing episodes per final policy for 5 independent training runs.

