# OpenReview forum: "Benchmarking Offline Reinforcement Learning on Real-Robot Hardware"
_ICLR.cc/2023/Conference — ICLR 2023 notable top 25%_

### Official Review · Reviewer_KUtS · 2022-10-24

**Confidence:** 4
**Correctness:** 4
**Technical Novelty And Significance:** 3
**Empirical Novelty And Significance:** 4
**Recommendation:** 8

**Clarity, Quality, Novelty And Reproducibility:**

The paper is very clear and well-written, and provides appropriate references.
The provided benchmark and algorithm’s analysis are both high quality.
The possibility of evaluating ORL algorithms  in a  real-world robotic system is novel.
The provided benchmark will allow to have reproducible results of the application of ORL algorithms.

**Details Of Ethics Concerns:**

No concerns

**Strength And Weaknesses:**

The benchmark of offline Reinforcement Learning (ORL) algorithms is a very relevant issue for comparing them and studying their applicability. The paper addresses the key issue of providing valuable data for comparing ORL algorithms, but also the possibility to execute the learned policies on a real-world robotic system. This is very relevant because the comparison of this kind of algorithms requires evaluating the so-called "distribution shift", which from my point of view requires experimenting in the real-world. This is the main paper's strength, in addition of being very clear and well written.

I don’t see any relevant weaknesses.

**Summary Of The Paper:**

This paper proposes a benchmark for offline Reinforcement Learning algorithms, which includes simulated data, real-world data and the possibility to execute the learned policies on a real-world robotic system. The robot application is dexterous manipulation, the specific tasks are Push and Lift, and the robotic platform TriFinger. Moreover, the paper presents a benchmark of prominent open-sourced offline RL algorithms (CRR. AWAC, CQL, and IQL), and a detailed analysis of the obtained results.

**Summary Of The Review:**

This is a good paper. The proposed benchmark of ORL algorithms is very valuable and will allow the RL community comparing ORL algorithms and studying their applicability.

---

> ### Author Response · Authors · 2022-11-17
> **Response to review**
>
> Thank you for your helpful review!
>
> To further increase the diversity of the proposed datasets, we added four new ‘Mixed’ datasets to our revision which are collected with a range of training checkpoints. This makes it harder for behavior cloning like algorithms to learn from the data. The Mixed datasets consequently proved quite challenging for the considered algorithms despite containing a large portion of successful episodes.

---

### Official Review · Reviewer_zztz · 2022-10-24

**Confidence:** 2
**Correctness:** 4
**Technical Novelty And Significance:** 1
**Empirical Novelty And Significance:** 4
**Recommendation:** 8

**Clarity, Quality, Novelty And Reproducibility:**

The paper is well written and easy to understand. It is also highly reproducible, given that the datasets really are published with the paper.

The novelty is purely in providing an openly available benchmark for offline RL algorithms on real-robot hardware.

**Strength And Weaknesses:**

Strengths:
- The experimental evaluation is extensive and very detailed.
- The reproducibility of the results seems very good. In particular the authors announce that they will publish the datasets and they will offer access to the real-robot platform for validation

Weaknesses:
- This is a pure benchmark paper. There is no technical contribution in terms of novel models or methods.

**Summary Of The Paper:**

The authors propose a novel benchmark for offline RL algorithms on real-robot hardware. The authors present two tasks, a pushing and a lifting tasks, which can be performed on the TriFinger platform. For each task an expert policy is learned on a simulator with extensive domain randomization. Afterwards the expert policy is used to gather two datasets, one from a simulated environment and one from a real-robot platform.
The authors then evaluate several SOTA offline RL algorithms on all four datasets and test the effect of including "weak trajectories" as well as reducing the amount of available expert data.

**Summary Of The Review:**

Since this is a pure benchmark paper the requirements concerning an extensive experimental validation are very high. In my opinion though they are fulfilled in this case. Additionally, I find that there is a considerable need for a real-robot benchmark in the offline RL community, which this paper contributes with great reproducibility. Therefore I vote to accept this paper.

---

> ### Author Response · Authors · 2022-11-17
> **Response to review**
>
> Thank you for your thorough review! We hope that the datasets we propose together with the open access to the TriFinger cluster will help the community to reliably measure progress in offline RL on real-robot hardware.
>
> To further increase the diversity of the datasets we propose, we added four new ‘Mixed’ datasets to our revision which are collected with a range of training checkpoints. This makes it harder for behavior cloning like algorithms to learn from the data. The Mixed datasets consequently proved quite challenging for the considered algorithms despite containing a large portion of successful episodes.

---

### Official Review · Reviewer_RMbA · 2022-10-25

**Confidence:** 5
**Correctness:** 4
**Technical Novelty And Significance:** 2
**Empirical Novelty And Significance:** 2
**Recommendation:** 6

**Clarity, Quality, Novelty And Reproducibility:**

Clarity: High. It is very clear what is being done in this work and the explanation and structure of the work relay it well.

Quality: Medium. Since this work introduces only two tasks on only one platform it limits it's applicability and distanciates from the claimed goal of being a "benchmark of RL on real robot hardware".

Novelty: Low. But that should not be taken as a critique, as this paper is not about being novel, it is about aggregating existing things into a useful package.

Reproducibility: High.

**Strength And Weaknesses:**

The paper reads well and leaves no confusion about what was done, how and why.

The purpose of this work is simple and is executed well, so that leaves little room for weaknesses as such. However what I would bring forward is the fact that in here we only deal with two datasets on only one platform. This set of tasks covers only a small portion of possible real-robot hardware tasks we (as a robot learning community) would like to benchmark our algorithms on. And within the subset of tasks that this work does address (dexterous manipulation) only a small set of challenges are presented.

For related work - here is another real robot platform that provides remote access for RL training https://arxiv.org/abs/1910.08639

**Summary Of The Paper:**

The authors introduce a real robot RL benchmarking platform for dexterous manipulation and, as a main contribution of this paper, collect two offline RL datasets and benchmark current popular Offline RL algorithms on these two datasets. This provides a point of reference to track future progress in Offline RL algorithms on these tasks.

**Summary Of The Review:**

I think for a field such as robotics having a set of well-established benchmarks is important, especially when we are talking about real world robotics and not sims. Currently each lab doing real-robot RL has its own hardware and RL problem definitions, which makes is hard to track progress in the similar manner to how we were able to track progress on Atari, MuJoCo, ImageNet etc. On one hand this work does not offer any novel scientific contributions, but on the other hand it also isn't the goal here. The goal is to get as many people as possible aware of the benchmark. Which is kind of the purpose of conferences as such - to make the community aware of things. With all that in mind I leaving a recommendation of "weak accept". In order to make a stronger recommendation the range of tasks would have to be wider and more diverse.

---

> ### Author Response · Authors · 2022-11-17
> **Response to review**
>
> Thank you for your constructive feedback!
>
> > However what I would bring forward is the fact that in here we only deal with two datasets on only one platform.
>
> We would like to point out that we benchmarked on 13 datasets in our original submission (two tasks, each in simulation and on the real robot with three dataset types plus a smoothed expert for lifting). While there are overlaps between the datasets because the Half-Expert variants are contained in their Weak&Expert counterparts (to isolate the effect of adding suboptimal trajectories), we believe that we offer variety in terms of task difficulty and dataset quality. To further increase the diversity of the proposed datasets, we included four additional datasets in our revision. These ‘Mixed’ datasets were recorded with a range of training checkpoints and have proved quite challenging for the benchmarked algorithms despite containing a large portion of successful trajectories.
>
> > This set of tasks covers only a small portion of possible real-robot hardware tasks we (as a robot learning community) would like to benchmark our algorithms on. And within the subset of tasks that this work does address (dexterous manipulation) only a small set of challenges are presented.
>
> We agree that more open access real-world robotics environments are needed for a more complete set of benchmarking tasks for the robot learning community. This is, however, beyond the scope of this work as the associated costs require the community as a whole to work towards this goal.
>
> > For related work - here is another real robot platform that provides remote access for RL training https://arxiv.org/abs/1910.08639
>
> Thank you for pointing this project out. We have added OffWorld gym to the related work section in our revision.

---

### Official Review · Reviewer_Yamp · 2022-10-26

**Confidence:** 4
**Clarity, Quality, Novelty And Reproducibility:** Paper is clear, novel and of good qua…
**Correctness:** 4
**Technical Novelty And Significance:** 3
**Empirical Novelty And Significance:** 3
**Recommendation:** 6

**Strength And Weaknesses:**

Strengths:

- I think this paper aims to solve a very important problem in robotics, which is the lack of proper comparisons and benchmarks.
- This paper presents an easy to use benchmarks with multiple tasks and multiple baselines
- The experiments conducted are thorough and insightful
- The paper is well written and very clear
- To my knowledge, no other such offline RL for robotics benchmarks exist that can allow for running the hardware directly

Weaknesses:

- While I think the experiments are interesting, implementation makes a big difference in the performance of RL algorithms. It would be good to include details about how these approaches are implemented should be included in Section 3.

- Additionally, more information should be included on the standardization protocol i.e., how each type of dataset is created exactly, and how to add other types. In general, it would be good to see more types of datasets with a visualization of the distributions of each.



**Summary Of The Paper:**

This paper presents a new benchmark for Offline Reinforcement Learning (RL). The benchmark contains two tasks, pushing a block and lifting it to a desired position. Rewards for each task are based on the object's pose (and distance to the goal pose). The hardware used is the TriFinger robot, available as a cluster. There is a pybullet simulation component to this benchmark as well. The datasets for offline RL have a mixture of expert, weaker and random policies, similar to other offline RL benchmarks. The paper shows evaluation of many SOTA offline RL benchmarks such as CQL, CRR, AWAC and IQL as well as behavior cloning.

**Summary Of The Review:**

Overall, I think this paper tackles an important problem of benchmarking in robotics and presents a good and useful solution to it. The benchmarks seems like it is ready for public use, and would be a benefit to the community. There are details on the exact implementations and setup of the benchmark missing, and adding the ability for users to do even more types of analysis and ablations would be interesting.

---

> ### Author Response · Authors · 2022-11-17
> **Response to review**
>
> Thank you for your helpful feedback!
>
> > While I think the experiments are interesting, implementation makes a big difference in the performance of RL algorithms. It would be good to include details about how these approaches are implemented should be included in Section 3.
>
> We use the implementations from the open-source library d3rlpy for offline RL. We added the URLs for the [code (available on GitHub)](https://github.com/takuseno/d3rlpy) and the [documentation](https://d3rlpy.readthedocs.io/) in Appendix C.2 of our revision and also specify the version of d3rlpy we used for our experiments. We omitted such details from the main text due to a lack of space. [1] moreover benchmarks the d3rlpy implementations on data from the standard MuJoCo gym environments and provides a point of reference on how well the implementations work.
>
> > Additionally, more information should be included on the standardization protocol i.e., how each type of dataset is created exactly
>
> Thank you for pointing this out. We have added more details on data collection and the choice of policies in Section B.1 in the appendix.
>
> > how to add other types [of datasets]
>
> We will publish the expert policies we used for data collection alongside the datasets and will provide a script for generating new datasets from the logs of jobs executed on the TriFinger cluster. These datasets can then simply be added in a fork of the dataset repository (and potentially to the original repository via a pull request).
>
> > In general, it would be good to see more types of datasets with a visualization of the distributions of each.
>
> To further increase the diversity of the proposed datasets, we added four new ‘Mixed’ datasets to our revision which are collected with a range of training checkpoints. As there is no single expert but a sequence of increasingly capable policies, it is difficult for behavior cloning like algorithms to learn from this data. The Mixed datasets consequently proved quite challenging for the considered algorithms despite containing a large portion of successful episodes.
>
> We  have furthermore added more plots of the return distribution in the real datasets to Figure S8. Please let us know if you had a different distribution in mind.
>
> [1] Takuma Seno and Michita Imai. d3rlpy: An offline deep reinforcement learning library. CoRR,
> abs/2111.03788, 2021. URL https://arxiv.org/abs/2111.03788.

---

### Decision · Program_Chairs · 2023-01-20

**Decision:**

Accept: notable-top-25%

**Justification For Why Not Higher Score:**

It is unclear if the benchmark for robotics reflects real-world problems in robotics.

**Justification For Why Not Lower Score:**

The paper is quite strong and does a thorough experimental study for benchmarking offline robotics.

**Metareview: Summary, Strengths And Weaknesses:**

This work presents a benchmarking platform for offline RL on real-world robotics tasks. This paper has received unanimously strong reviews. The proposed datasets, along with benchmark numbers will be valuable to the offline robotics community. Perhaps the most relevant weakness is if the benchmark actually reflects problems in robotics. If not, the results produced on the benchmark may not transfer to real-world problems. Nevertheless, this work is a step in a right direction and will spur better benchmarking.

**Note From Pc:**

if the above contains the word "oral" or "spotlight" please see: "oral" presentation means -> notable-top-5% and "spotlight" means -> notable-top-25%. As stated in our emails, we are disassociating presentation type from AC recommendations